# The NuRD complex cooperates with SALL4 to orchestrate reprogramming

Bo Wang[1,2,3,4,10], Chen Li[3,5,6,10], Jin Ming[1,3,5,6,10], Linlin Wu[1], Shicai Fang[3,5,7], Yi Huang[3,5,7], Lihui Lin[2,3,5,8], He Liu[2,3,5], Junqi Kuang[1,3,5,8], Chengchen Zhao[1], Xingnan Huang[1], Huijian Feng[2,3,5,6], Jing Guo[3,5], Xuejie Yang[3,5], Liman Guo[2], Xiaofei Zhang[2,3,5], Jiekai Chen[2,3,5], Jing Liu[2,3,5], Ping Zhu[4,9] ✉ & Duanqing Pei[1] ✉

Cell fate decision involves rewiring of the genome, but remains poorly understood at the chromatin level. Here, we report that chromatin remodeling complex NuRD participates in closing open chromatin in the early phase of somatic reprogramming. Sall4, Jdp2, Glis1 and Esrrb can reprogram MEFs to iPSCs efficiently, but only Sall4 is indispensable capable of recruiting endogenous components of NuRD. Yet knocking down NuRD components only reduces reprogramming modestly, in contrast to disrupting the known Sall4-NuRD interaction by mutating or deleting the NuRD interacting motif at its N-terminus that renders Sall4 inept to reprogram. Remarkably, these defects can be partially rescued by grafting NuRD interacting motif onto Jdp2. Further analysis of chromatin accessibility dynamics demonstrates that the Sall4-NuRD axis plays a critical role in closing the open chromatin in the early phase of reprogramming. Among the chromatin loci closed by Sall4-NuRD encode genes resistant to reprogramming. These results identify a previously unrecognized role of NuRD in reprogramming, and may further illuminate chromatin closing as a critical step in cell fate control.

Pluripotent stem cells (PSC) can be derived from the inner cell mass of blastocyst-stage embryos[1] or induced from somatic cells using a defined cocktail of factors such as Oct4, Sox2, Klf4, and c-Myc (OSKM)[2]. Induced pluripotent stem cells (iPSC) are functionally indistinguishable from embryonic stem cells (ESC) and exhibit remarkable developmental plasticity, capable of giving rise to all cell types of an organism except for extraembryonic tissues. The establishment and maintenance of PSC is attributed to their unique chromatin structure and the transcriptional regulatory network governing by core transcription factors such as Oct4, Sox2, Nanog, Esrrb, and Sall4[3].

Reprogramming somatic cells to pluripotent stem cells or a particular cell type of interest represents a new paradigm for both basic biological sciences and translational research[2,4–8]. Classic Yamanaka reprogramming (OSKM) has provided conceptual understanding

[1]Laboratory of Cell Fate Control, School of Life Sciences, Westlake University, Hangzhou, China. [2]Center for Cell Lineage and Atlas (CCLA), Bioland Laboratory, Guangzhou Regenerative Medicine and Health Guangdong Laboratory, Guangzhou, China. [3]CAS Key Laboratory of Regenerative Biology, South China Institutes for Stem Cell Biology and Regenerative Medicine, Guangzhou Institutes of Biomedicine and Health, Chinese Academy of Sciences, Guangzhou, China. [4]Guangdong Cardiovascular Institute, Guangdong Provincial People's Hospital (Guangdong Academy of Medical Sciences), Southern Medical University, Guangzhou, China. [5]Guangdong Provincial Key Laboratory of Stem Cell and Regenerative Medicine, South China Stem Cell and Regenerative Medicine, Guangzhou Institutes of Biomedicine and Health, Chinese Academic of Sciences, Guangzhou, China. [6]University of Chinese Academy of Sciences, Beijing, China. [7]Joint School of Life Science, Guangzhou Medical University-Guangzhou Institutes of Biomedicine and Health, Chinese Academy of Sciences, Guangzhou, China. [8]Guangzhou Branch of the Supercomputing Center of Chinese Academy of Sciences, Guangzhou, China. [9]Guangdong Provincial Key Laboratory of Pathogenesis, Targeted Prevention and Treatment of Heart Disease, Guangzhou Key Laboratory of Cardiac Pathogenesis and Prevention, Guangzhou, China. [10]These authors contributed equally: Bo Wang, Chen Li, Jin Ming. ✉e-mail: tanganqier@163.com; peiduanqing@westlake.edu.cn

of the reprogramming process through various experimental conditions[9,10]. The fact that reprogramming can be achieved through non-Yamanaka methods such as combinations of factors or chemicals suggests that reprogramming can start at divergent points or pathways[11–16], but eventually converge to a common path towards pluripotency, thus, unifying the classic Yamanaka system with all these alternatives. This unified theory is consistent with earlier reports on mesenchymal-epithelial transition (MET), barriers, and epigenetic regulations[17–19]. Extensively studied chromatin-based barriers to transcription factor mediated reprogramming include repressive chromatin factors and factors associated with active transcription[20]. Repressive factors are associated with actively transcribed loci in somatic cells, such as H3K4/36/79 methylation[21–23], SUMOylation[24], Facilitates Chromatin Transcription (FACT)[25], and Poly (ADP-Ribose) Polymerase 1 (PARP1)[26]. Genetic screening identified numerous barriers with repressive functions that act on pluripotent-associated genes, including DNA Methyltransferases (DNMT)[27], Nucleosome Remodeling and histone Deacetylation (NuRD)[28], histone deacetylases (HDACs)[29], H3K9 methylation[30], and chromatin assembly factor 1 (CAF1)[31]. Among the barriers identified, the NuRD complex appears to be quite controversial[32–34]. The NuRD complex consists of seven core components, each with multiple paralogues, including HDAC1/2, CHD3/4, RbAp46/48, MBD2/3, GATAD2a/b, and MTA1/2/3. These components perform various functions such as histone deacetylation, ATP-dependent chromatin remodeling, histone chaperoning, CpG-binding, DNA-binding, and transcriptional regulation[35]. Near 100% reprogramming has been achieved by deleting MBD3, a component of NuRD[28]. This claim was disputed by others showing that the role of NuRD is context dependent, e.g., being a positive regulator in one reprogramming system while negative in others[32,36,37].

The generation of iPSCs has been known to involve a direct interaction between reprogramming factors and chromatin regulators in order to overcome epigenetic barriers. According to recent research, Oct4 has the ability to interact with SWI/SNF complex compounds, which enhances reprogramming by facilitating OCT4 binding to target promoters[38]. Additionally, the interaction between Nanog and Sin3a allows for a synergistic transcriptional program on pluripotent gene activation and the overcoming of reprogramming barriers[39]. Of note, Sall4 is crucial for the early development of embryos, and mice without Sall4 cannot survive beyond E6.5[40–42]. Sall4 has been shown to have a positive role in generating iPSCs from somatic cells and can replace OSKM when overexpressed with other factors during reprogramming[43,44]. The establishment and maintenance of pluripotency is dependent on Sall4's interaction with various proteins. It has a conserved N-terminal 12 amino acid (N12-aa) that interacts with the NuRD complex, which is also found in Sall1 and Sall3[45]. Additionally, Sall4 plays a crucial role in maintaining the undifferentiated state of PSC by binding to Oct4, Sox2, and Nanog. It acts as a transcription repressor by interacting with LSD1 and DNMTs. Furthermore, a collaborative role between Sall4 and TET proteins was discovered in stepwise oxidation of DNA methylation in ESC[46]. Among above protein-partners, the specific protein-partners that interact with Sall4 to define its molecular function remain unknown. While the inhibitory role of NuRD in reprogramming is established in lots of studies, Sall4 has been found to facilitate reprogramming. However, the mechanism of Sall4 integrating with the transcriptional repressor, NuRD complex, to coordinately regulate pluripotency establishment remain further investigation.

Here, we present evidence that the NuRD complex play a critical role in efficient reprogramming driven by Jdp2, Glis1, Esrrb and Sall4 (JGES). Sall4 recruits NuRD complex to open chromatin in MEFs to ensure the closure of somatic loci. This recruitment is dependent on the N-terminal motif of Sall4 and can be transferred to an unrelated factor such as Jdp2. Our results suggest that NuRD complex plays a positive role in early phase of reprogramming by closing chromatin in MEFs.

## Results

### JGES reprogramming through a Sall4-NuRD axis

The 7 F (Jdp2, Esrrb, Sall4, Nanog, Glis1, Kdm2b, Mkk6) system we described previously achieved efficiency better than the classic OSKM system, but remains cumbersome in mechanistic studies[47]. To this end, we embarked on an optimization process that eventually gives rise to a 4 F system, Jdp2, Glis1, Esrrb and Sall4 (JGES). We first optimized the culture medium, by starting out with iCD1 (iPS Chemically Defined medium1) and testing each component through dropout experiments (Supplementary Fig. 1a), that identifies LiCl being detrimental and confirms vitamin C and LIF critical (Supplementary Fig. 1b). Then, we performed a screen for chemicals that can boost 7 F reprogramming (Supplementary Fig. 1c) and identified two additional chemicals, GSK-LSD1-2HCL and SGC0946 capable of further improving 7 F reprogramming (Supplementary data 1). Along with ROCK inhibitor previously shown to enhance 7 F reprogramming, we formulated iCD3 (iPS Chemically Defined medium3) (Supplementary Fig. 1d) and show that it improves 7 F reprogramming by about 100%, generating 12–13 colonies per 150 MEF cells, ~8%, in 7 days (Supplementary Fig. 1e, f). To reduce the number of factors in 7 F, we performed a dropout experiment for each factor and show that each factor appears to contribute significantly to 7 F reprogramming in iCD3, although with various degree of impact (Supplementary Fig. 1g). We then show that JGES reprogram MEFs to iPSCs at the efficiency comparable to 7 F under iCD1 (Supplementary Fig. 1h, i). We then picked colonies from JGES to establish iPSC clones and tested their transgene integration. As shown in Supplementary Fig. 1j, iPSC clones from JGES contain those 4 genes, without the classic Yamanaka factor OSKM nor Nanog, Kdm2b and Mkk6 from 7 F (Supplementary data 2, 3). The JGES clones can be passed stably (Supplementary Fig. 1k), possess normal karyotypes (Supplementary Fig. 1l) and can generate chimera with blastocysts injection that can undergo germ-line transmission (Supplementary Fig. 1m). We can also demonstrate that Oct4-GFP positive clones picked at Day 7 can give rise to chimera with germline transmission without further passaging (Supplementary Fig. 1n). Thus, JGES, like the classic OSKM, may be suitable for mechanistic studies.

We ask whether each of JGES factors contributes to reprogramming equally as OSKM. We assessed this by performing dropout experiments and show that, surprisingly, while dropping J, G, E individually weakens reprogramming significantly, but Sall4 appears to be more important than the other 3 as its removal renders reprogramming close to 0 (Fig. 1a). Consistently, when we looked at the bulk RNAseq datasets from the drop-out experiments in Fig. 1a, JGE is the only one with a divergence towards the left as all others move to the right towards ESCs along the PC1 axis (Fig. 1b). Based on these results, we decided to focus on Sall4 for further mechanistic analysis.

We hypothesize that Sall4 must engage a critical cellular component to overcome a major barrier during reprogramming as SALL4 has been shown to interact with many proteins. To identify such partner(s), we performed IP-Mass on MEFs infected with JGES with anti-SALL4 antibody and show by pairwise comparison that SALL4 co-purifies with canonical subunits of the NuRD complex (Fig.1c, Supplementary data 4), suggesting that collaborative interactions exist between SALL4 and the NuRD complex.

NuRD has been implicated in reprogramming in previous studies[32–34,48,49], but with mixed results and divergent mechanistic explanations. To investigate its role in JGES reprogramming, we performed knock-down experiments on NuRD subunits and show that among all the 13 canonical subunits, knocking-down Gatad2b/2a and Chd4 significantly reduce *Oct4*-GFP[+] colonies (Fig.1d, Supplementary Fig. 2a, Supplementary data 5, 6). Principal component analysis (PCA) shows that there is a delay of transition from somatic state to pluripotent state with shGatad2b (Yellow) or shChd4 (Blue) compared to control (Green). Though one of the shChd4 samples on day7 looks like

shLuciferase, another shChd4 sample and two shGatad2b samples on day 7 show further distance to ESCs than shLuciferase at day 5. Compared to shChd4 samples at day7, shGatad2b samples at day7 are much farther away from ES cells, suggesting that depleting Gatad2b in MEFs restrains the conversion of MEFs to the pluripotent state (Fig.1e). Intriguingly, comparative transcriptome analysis revealed that genes with altered expression in shGatad2b or shChd4 are associated with response to interferon-beta, striated muscle contraction, extracellular matrix organization (Fig.1f). Additionally, we found that many genes like Rasa3, Bicc1, and Tmem98 failed to be downregulated when Gatad2b was knocked down (Supplementary Fig. 2b). In combination,

these results suggest that Sall4 need assistants from subunits of NuRD complex during JGES reprogramming.

## NuRD mediates chromatin closing of somatic loci

Given the mixed roles of NuRD reported earlier in reprogramming studies, we wish to resolve its role in JGES reprogramming by ATAC-seq to analyze the chromatin accessibility dynamics (CAD). Consistent with the reduction of reprogramming efficiency, we observed that shGatad2b and shChd4 impact CAD quite dramatically, altering CO (close to open) and OC (open to close) peaks (Fig.1g). For example, the number of CO1, CO2, and CO3 peaks are much higher in shGatad2b

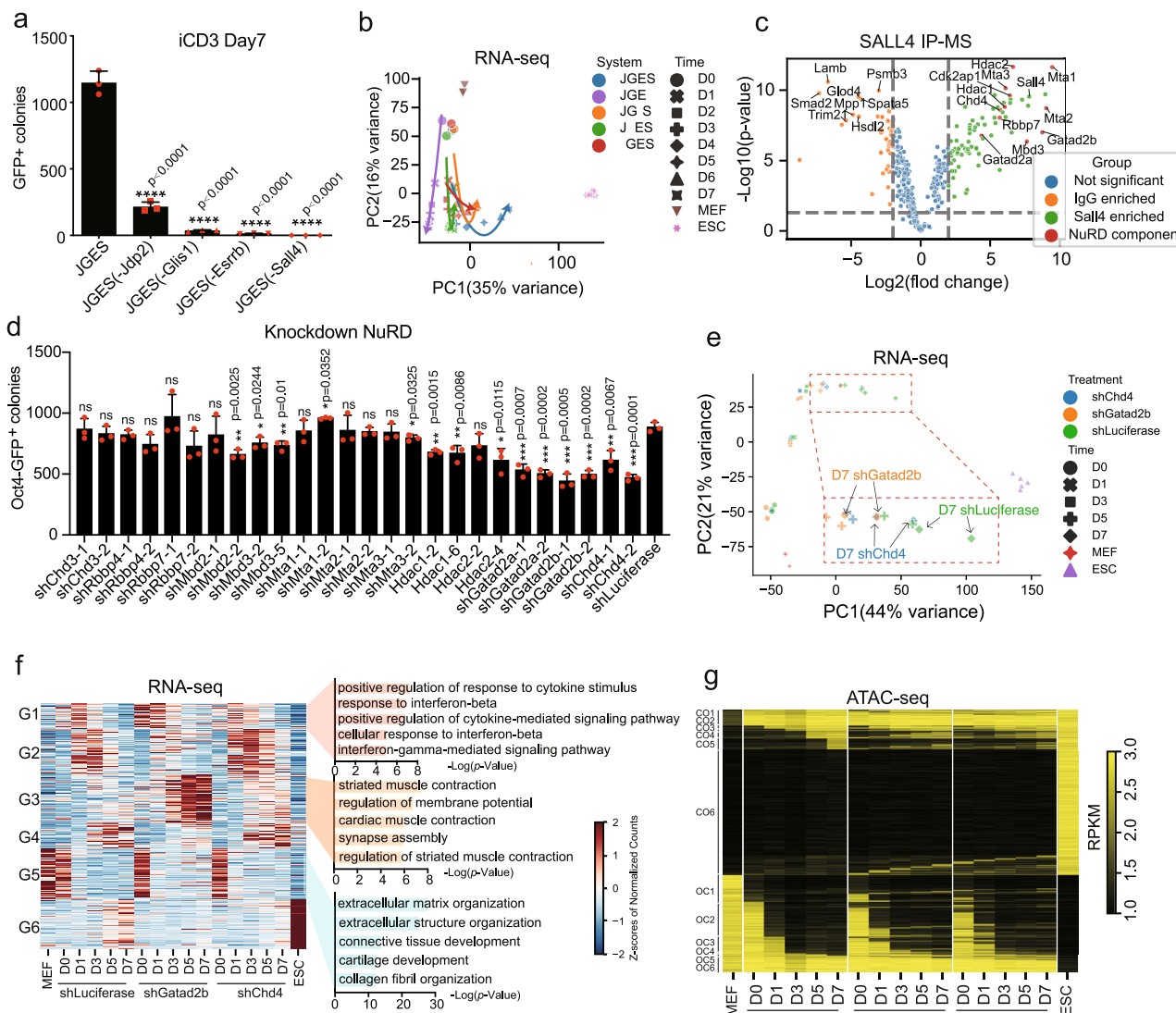

**Fig. 1 | Knockdown NuRD complex subunits compromised JGES mediated iPS induction. a** Bar plot of numbers for Oct4 GFP positive iPS colonies of MEF cells reprogrammed by JGES or individual factor dropout in iCD3 medium for 7 days. Data are mean ± s.d., two-sided, unpaired *t*-test; *n* = 3 independent experiments, ****$p$ < 0.0001. **b** Principal-component analysis (PCA) of RNA-seq for MEF, JGES, individual factor dropout reprogramming and ESC samples. Solid line with arrows represents one reprogramming sample at sequential time. **c** Volcano plots of SALL4 significantly enriched proteins of reprogramming samples at day3 performed in triplicate for SALL4 and IgG pull down followed by MS analysis. IP-MS experiments were performed in triplicates and a two-sided *t*-test was applied. *P* = 0.05 and fold change = 1.5 were used as threshold. **d** Bar plot of numbers for Oct4 GFP positive iPS colonies by knockdown NuRD subunits. Data are mean ± s.d., two-sided, unpaired t-test; *n* = 3 independent experiments, *$p$ < 0.05, **$p$ < 0.01,

***$p$ < 0.001. **e** PCA analysis of RNA-seq for the roadmap of MEF cell towards ESC by knockdown Gatad2b or Chd4. A zoomed-in snapshot of all day5 and day7 samples was placed on the button of the PCA map. **f** Heat map showing the differential expressing genes in reprogramming with knocking down Gatad2b and Chd4 in JGES. Genes were clustered into 6 groups according to change pattern. Bar plot showing the gene ontology (GO) analysis of the category. G1-6 indicates group1–6. The data in the heat map were clustered using minisom (a minimalistic and Numpy based implementation of the Self Organizing Maps (SOM)). The *p*-value in GO results were calculated by hypergeometric test. **g** Heat map showing the change of chromatin accessibility during reprogramming in knockdown Luciferase, Gatad2b or Chd4 in the reprogramming of JGES. Regions were divided into open-close (OC) state change or close-open (OC) state change with the sequence of occurrence in control. Source data related to Fig. 1a, d are provided as a Source Data file.

and shChd4 than in shLuciferase, suggesting that many loci are opened improperly in shGatad2b and shChd4 at reprogramming day0, day1 and day3. The number of OC4 and OC5 peaks is much higher in shGatad2b and shChd4 than in shLuciferase, indicating that more loci fail to be closed in shGatad2b and shChd4 at reprogramming day3 and day5 than in shLuciferase (Supplementary Fig. 2c). Specifically, only 45.45% and 24.96% loci in CO2 of shGatad2b and shChd4 overlap with those of shLuciferase, 59.06% and 68.12% loci in OC1 change as expected, respectively (Supplementary Fig. 2d, e). We calculated the number of genes that gene body or promoter was located in the ATAC-seq peaks region (Supplementary Fig. 2f, g). Consistent with chromatin accessibility results, hundreds of additional genes were enriched by Gatad2b and Chd4 knockdown when compared to the control. Interestingly, when compared to the common and shLuciferase parts, fewer genes were enriched from OC1 and OC2 but more genes were found in OC3-OC6 by Gatad2b knockdown. Similar results could be found from the Chd4 knockdown reprogramming (Supplementary Fig. 2f, g). These results indicated that there is a delay of lots of gene regions becoming inaccessible during reprogramming by Gatad2b/Chd4 knockdown. Besides, when analyzing the Sall4, Gatad2b occupancy and H3K27ac modification at OC regions, Sall4 and Gatad2b displayed a higher binding density at shLuciferase than shGatad2b and shChd4 specific regions. However, an increased H3K27ac signal was found by Gatad2b or Chd4 knockdown (Supplementary Fig. 2h, i). These data support that the NuRD complex is involved in the inactivation of the somatic program early in JGES reprogramming. To further analyze the dynamics of progressively closing of chromatin loci, we defined gradually closed regions (GCRs) by subtracting ATAC-seq signals between adjacent stages below a threshold which was set at 0.05 multiplied by the range of the normalized ATAC-seq signal. Knocking-down either Chd4 or Gatad2b appears to slow down closing of somatic loci (Supplementary Fig. 2j), as measured by the normalized signal intensity of the gradually close region (GCR). The GCRs are dominated by transcription factor motifs for AP-1, ETS, and TEAD family genes such as JUNB, ETV1, and TEAD1 (Supplementary Fig. 2k).

We then approached the Sall4-NuRD axis through their known interactions through the N-terminal 12 residues[50] by mutating individual residues (Fig. 2a) and show that Sall4[P9A], Sall4[R3A], Sall4[R4A], and Sall4[K5A] render it no longer able to reprogram (Fig. 2b). We then focused on Sall4[K5A] and show that this mutant fails to interact with NuRD through IP-Mass or CO-IP (Fig. 2c, d, processed data provided in Supplementary data 1). Consistently, this mutant also fails to mediate the requisite transcriptomic and chromatin reprogramming as measured by RNA- and ATAC-seq, respectively (Supplementary Fig. 3a, b).

Unlike NuRD, we also show, through unbiased analysis of the proteomics data, that Sall4[K5A] and Sall4[WT] also share many protein partners (Fig. 2e, detail provided in supplementary data1). Gene ontology analysis show that the common proteins are associated with DNA repair (Supplementary Fig. 3c). To validate the function of those proteins, a decrease in reprogramming efficiency was observed when we knocked down Hmbox1, Tfam, Parp1, Lig3, and Kpna4 by shRNA (Supplementary Fig. 3d, e), suggesting that Sall4 engages many partners besides NuRD to facilitate reprogramming. Yet since Sall4[K5A] is almost totally ineffective, we conclude that the NuRD-Sall4 axis may play a dominant role in JGES reprogramming.

The multiple zinc fingers of Sall4 have unique roles in regulating downstream target gene expression by its zinc finger clusters (ZFC)[46,51,52]. To determine their contribution to JGES reprogramming, we made mutation or deletion as detailed in Supplementary Fig. 3f. Consistent with point mutation data described above, deleting N12 abolishes Sall4-dependent reprogramming, while other mutations have either no effect (ZF1) or limited impacts (C420A, ZFC2-4) (Supplementary Fig. 3g, h). confirming that Sall4 mediates reprogramming primarily through its N12 domain that engages NuRD and secondarily through ZFC2-4 that likely engages the above-mentioned factors

involved in DNA repairs. As such, we continued to focus on the Sall4-NuRD complex in this study.

Compared to the shChd4 and shGatad2b data in Fig.1d with ~ 50% reduction in iPSC colonies, Sall4[K5A], like Sall4[P9A], Sall4[R3A], and Sall4[R4A], on the other hand, has ~100% reduction (Fig.2b), offering the opportunity to assess the Sall4-NuRD axis in mediating chromatin closing. So, we performed ATAC-seq on MEFs undergoing reprogramming with Sall4[WT] vs Sall4[K5A] and compared the pattern of open and close chromatin. A total of 9344 ATAC-seq peaks can be classified into 6 clusters according to chromatin accessibility (Fig. 2f). Among the 6 clusters, more than 2/3 of regions are open in Sall4[K5A] but closed in Sall4[WT] (C4, $n = 4573$ and C6, $n = 1904$). Besides, there are two interesting clusters that exhibit a loss of accessibility in Sall4[K5A] but become accessible (C5) and inaccessible (C3) progressively in Sall4[WT], respectively. Examining the expression for genes whose promoter located within ATAC-seq peaks in each cluster indicates that patterns in transcription match those of chromatin accessibility (Fig. 2g). For regions that are more accessible in Sall4[WT], we observed a higher level of gene expression in Sall4[WT] than in Sall4[K5A] such as Mas1, Peg10, and Pkd1 (Supplementary Fig. 3i, j). In contrast, for regions in which accessibility is established in Sall4[K5A] but remains inaccessible in Sall4[WT], there was a more significant increase in gene expression in Sall4[K5A] than in Sall4[WT] such as Bicc1, Fmo1, and Sox5 (Supplementary Fig. 3i, j). Gene ontology analysis of genes associated with distinct clusters showed that the C4 and C6 loci correspond to those related to somatic cell maintenance and differentiation (e.g., regulation of mesenchymal stem cell differentiation), while the C5 loci are associated with cell cycle phase transition (Supplementary Fig. 3k).

Motif enrichment for each cluster shows that enriched motifs are quite different between Sall4[WT] and Sall4[K5A]. For example, motifs from ETS (ETS1) and HOMEBOX family (Lhx3, Lhx1, Dlx1, Dlx3) members are specifically enriched in C6 and C4, respectively. Motifs for ETS and FOX family (FoxK2, FoxO3) members are both found in C6 and C4. Moreover, motifs for TFs from the AP-1 family such as Fosl2, Fra1/2, c-Jun, and JunB are also present in C6 (Fig. 2h).These results are entirely consistent with our earlier findings that somatic gene loci enriched with somatic state specific TFs from AP-1 and ETS family members are barriers for reprogramming[10,53,54]. Several TFs have already been shown significantly inhibits iPSC induction such as FoxK2 and FoxO3[55]. Lhx3 and Dlx1/2 selectively drive fibroblast to distinct subtypes of neurons[6,56–58]. On the other hand, motif enrichment for C5 shows that pluripotent TFs such as OCT4/6, KLF4, and SOX17/21 are only found in Sall4 WT but not in K5A. These results suggest that the interaction between Sall4 and NuRD complex is required to reconfigure the chromatin architecture for reprogramming.

Consistent with the failure to close somatic loci, we show by RNA-seq that genes in G3 fail to be downregulated and are related to the MEF somatic state (extracellular matrix organization) (Fig. 2i). Furthermore, Sall4[K5A] appears to divert cell fate towards innate immunity such as interferon-beta and complement activation in G6 (Fig. 2i). Coincidentally, those GO terms observed in Sall4[K5A] groups can also be found after depletion of Gatad2b and Chd4 during JGES reprogramming (Fig. 1f). Consistently, by comparing the RNA-seq data in each group (Fig. 1f) with Sall4[K5A] upregulated or downregulated gene sets, we show statistically significant concordance between shGatad2b/Chd4 and Sall4[K5A] (Supplementary Fig. 3l). Together, these results suggest that Sall4-NuRD axis is important for closing somatic chromatin loci during reprogramming.

## Sall4 K5A mutation results less occupancy of Gatad2b and increased H3K27ac at somatic loci

To test whether failure to close somatic loci associated with Sall4[K5A] is dependent on histone modification, we performed CUT&Tag by anti-SALL4 and H3K27ac antibody on MEFs infected with Sall4[WT] and Sall4[K5A], undergoing reprogramming at day1. Quantification of SALL4

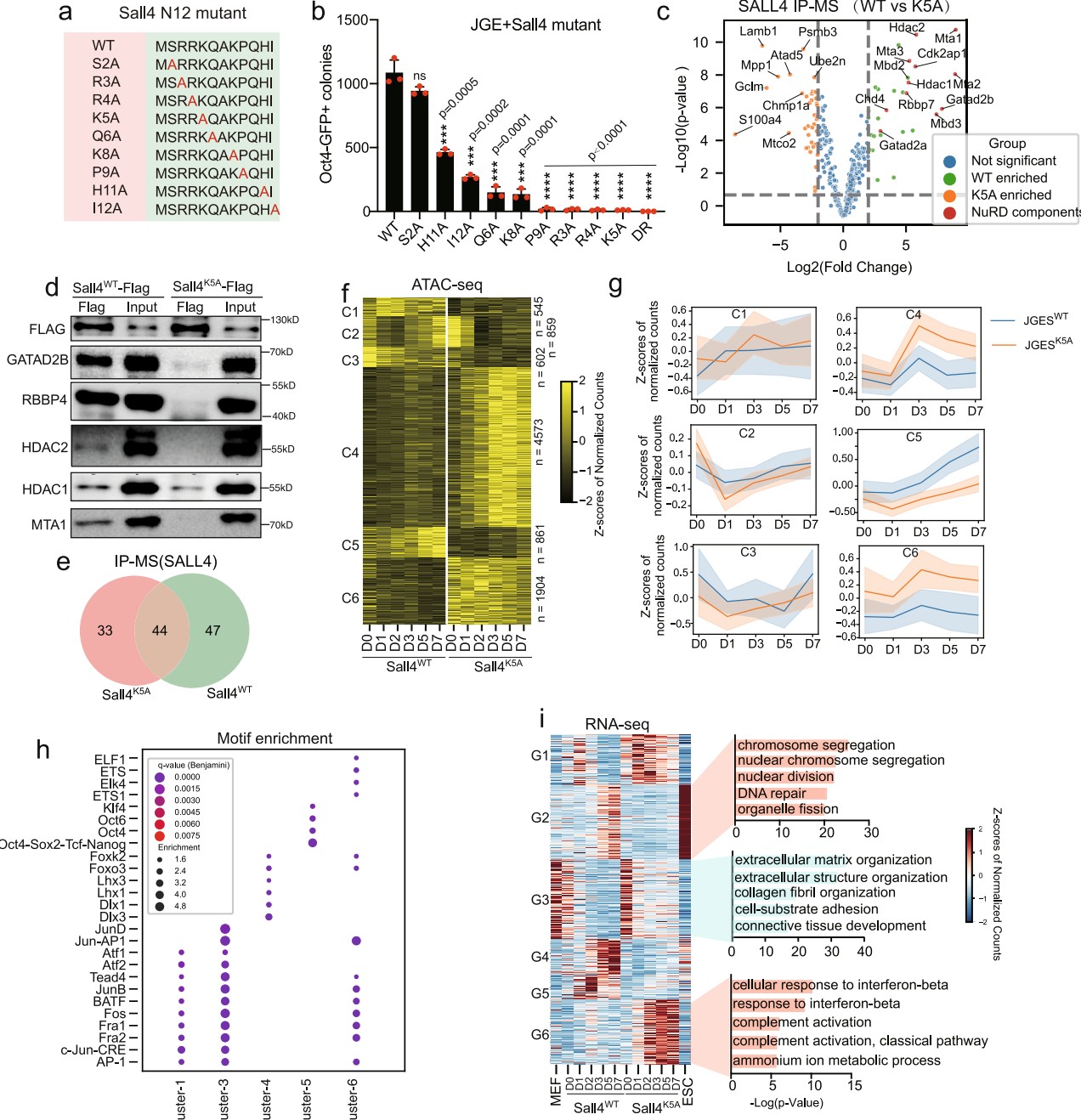

**Fig. 2 | Sall4 recruit NuRD complex to regulate chromatin accessibility by its N terminus 12 amino acid. a** Schematic of Sall4 N terminal amino acid point mutation. **b** Bar plot of numbers for Oct4 GFP positive iPS colonies induced by JGE plus Sall4 point mutation. Wild type Sall4 and DsRed indicated as positive and negative control, respectively. Data are mean ± s.d., two-sided, unpaired t test; $n = 3$ independent experiments, ***$p < 0.001$, ****$p < 0.0001$. Source data is provided as a Source Data file. **c** Volcano plots of enriched proteins by WT and K5A mutated SALL4 pull down followed by MS analysis. IP-MS experiments were performed in triplicates and a two-sided $t$ test was applied. $P = 0.05$ and fold change = 1.5 were used as threshold. **d** Flag tagged WT or K5A mut Sall4 were transfected into MEFs in combination with JGE for reprograming. The day1 cell lysates were immunoprecipitated (IP) with anti-FLAG antibody, followed by an immunoblot analysis by anti-GATAD2B, RBBP4, HDAC1/2, MTA1 antibody. **e** Venn shows numbers of protein common and specific purified by Sall4[WT] and Sall4[K5A]. **f** Heat map showing the

regions of differential chromatin accessibility during Sall4[WT] and Sall4[K5A] reprogramming. The number of peaks in each cluster was list in right side. **g** Line plot of gene expression during Sall4[WT] and Sall4[K5A] reprogramming, whose promoter are located in the region clustered in (**f**). The shaded area around the line represents the margin of error (95% confidence interval by using Bootstrap Method). **h** Enrichment of TF motifs in ATAC-seq cluster in (**f**). Point size represents the proportion of sequences in the cluster featuring the motif and red gradient the enrichment significance. **i** Heat map showing the differential expressing genes in Sall4[WT] and Sall4[K5A] reprogramming. Genes were clustered into 6 groups according to change pattern. Bar plot showing the gene ontology (GO) analysis of the category. G1-6 indicates group1-6. The data in the heat map were clustered using minisom (a minimalistic and Numpy based implementation of the Self Organizing Maps (SOM)). The p-value in GO results were calculated by hypergeometric test.

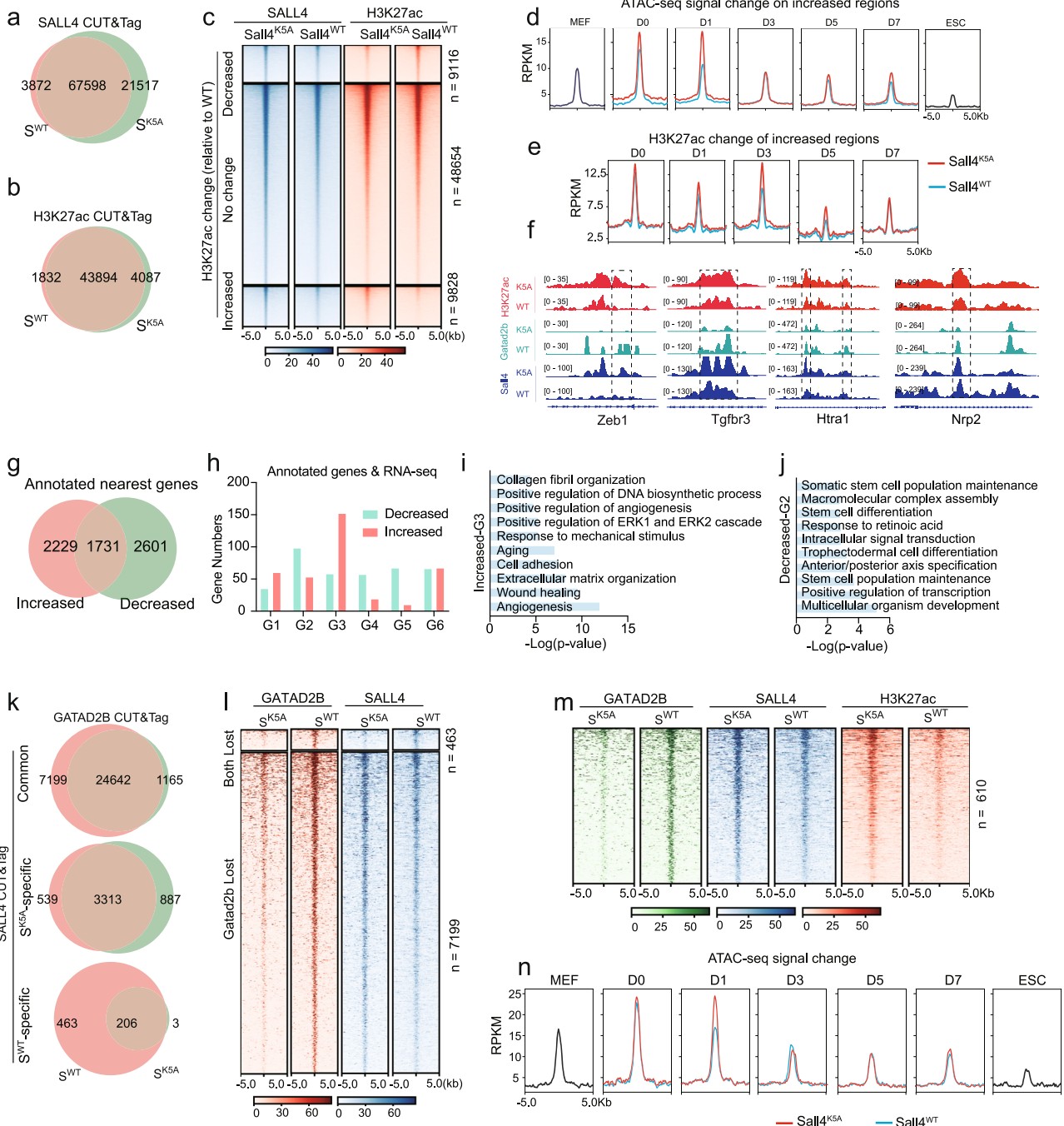

**Fig. 3 | SALL4-NuRD complex cooperate to close somatic chromatin. a, b** Venn diagram showing the overlap of CUT&Tag peaks. CUT&Tag was performed by SALL4 or H3K27ac antibody on day 1 in JGES$^{WT}$ or JGES$^{K5A}$ reprogramming. **c** Heat map illustrating the density of H3K27ac CUT&Tag reads on the Sall4-bound sites on day 1 in JGES$^{WT}$ or JGES$^{K5A}$ reprogramming. The increased/decreased region was defined by comparing the H3K27ac signal density. The peaks number listed in right. **d** Pileup of ATAC-seq signal of the increased regions in **c** for MEF, day0-7(JGE +Sall4$^{WT}$/Sall4$^{K5A}$ reprogramming), and ESC. **e** Pileup of H3K27ac signal for the increased regions in **c** during JGE+Sall4$^{WT}$/Sall4$^{K5A}$ reprogramming(day 0–7). **f** Selected genomic views of H3K27ac modification, Sall4 and Gatad2b binding sites for the indicated genes on day 1 in JGES$^{WT}$ or JGES$^{K5A}$ reprogramming. Zeb1(chr18:5,612,430-5,623,117); Tgfbr3 (chr5:107,164,974-107,169,528); Htra1(chr7:130,958,612-130,970,880); Nrp2(chr1:62,716,659- 62,725,726). **g** Venn diagrams showing the overlap of annotated nearest genes from increased and

decreased regions in **c**. **h** Histogram showing the overlap of genes between annotated nearest genes (Fig. 3g) and RNA-seq data (Fig. 2h). **i, j** Gene ontology analysis of the overlap genes between increased-G3 and decreased-G2. The *p*-value in GO results were calculated by hypergeometric test. **k** Venn diagram showing the Gatad2b's overlapping peaks on Sall4 differential binding sites in (**a**) on day1 of reprogramming. **l** Heat map illustrating the density of Gatad2b and Sall4 CUT&Tag reads on day1 of reprogramming. Both lost regions indicated the sites which Gatad2b and Sall4 only bind in JGES$^{WT}$ reprogramming. Gatad2b lost regions indicated the sites which Gatad2b only bind in JGES$^{WT}$ reprogramming while Sall4 bind in both WT and K5A reprograming. **m** Heat map illustrating the density of Gatad2b, Sall4 and H3K27ac CUT&Tag reads on day1 of reprogramming. The regions were the subset of Gatad2b lost in (Fig. 3l) with H3K27ac density increased in JGES$^{K5A}$ reprogramming. **n** Pileup of ATAC-seq signal of regions in (Fig. 3m) for MEF, day0- 7(JGE+Sall4$^{WT}$/Sall4$^{K5A}$ reprogramming) and ESC.

CUT&Tag signal showed that over 75% (67598) of Sall4$^{K5A}$ peaks overlap with those of Sall4$^{WT}$, indicating that Sall4$^{WT}$ and Sall4$^{K5A}$ occupy similar loci (Fig. 3a). When mapping the CUT&Tag dataset to global occupancy of H3K27ac, we identified a total of 48913 peaks and 88% (43894) peaks are enriched from both Sall4$^{WT}$ and Sall4$^{K5A}$ (Fig. 3b). So, we focus on the 67598 common target peaks from Sall4$^{WT}$ and Sall4$^{K5A}$ CUT&Tag data to evaluate the levels of H3K27ac. Compared to Sall4$^{WT}$ control, the H3K27ac density in Sall4$^{K5A}$ could be divided into three basic groups, declining (decreased) or increasing in intensity (increased), no change (Fig. 3c). Basically, in 72% (48654) of the gene loci, the level of H3K27ac remains similar regardless of WT or K5A. However, 14.5% (9828) of the loci have higher H3K27ac density in Sall4$^{K5A}$ reprogramming. For regions with elevated H3K27ac in Sall4$^{K5A}$, we observed a gradual loss of chromatin accessibility from day1 to the end of reprogramming (Fig. 3d). Of note, compared to Sall4$^{WT}$, not only the average level of H3K27ac but also chromatin accessibility shows much higher level in Sall4$^{K5A}$ samples (Fig. 3d, e). Consistent with this observation, many somatic gene loci engaged by Sall4 (*Zeb1, Tgfbr3, Htra1* and *Nrp2*) are associated with higher levels of H3K27ac in Sall4$^{K5A}$ (Fig. 3f). On the other hand, reduced level of H3K27ac and chromatin accessibility were found in Sall4$^{K5A}$ at several pluripotent gene loci (*Sox2, Tead4,* and *Wnt6*) (Supplementary Fig. 4a–c), suggesting that certain chromatin regions need be opened and activated during this process. Based on the same motif enrichment method mentioned above, motif enrichment in those three groups showed that motifs for somatic TFs, such as JunB, Fra1/2, Fosl2, and BATF are enriched from the no change group. Interestingly, much more significant enrichment of those motifs was observed in H3K27ac elevated group (Supplementary Fig. 4d), suggesting that without NuRD, Sall4$^{K5A}$ occupies loci with high H3K37ac, and consequently fails to close them properly.

To explore transcriptional changes regulated by Sall4, genes located within an increased or decreased region were considered as a putative annotated nearest gene (Fig. 3g). Integrating analysis with RNA-seq, we show that annotated nearest genes from increased and decreased regions correspond predominantly to RNA-seq group 3 and group2, respectively (Fig. 3h). Gene ontology analysis of G3 and G2 show that the increased-G3 correspond to genes related to somatic state (e.g., angiogenesis, wound healing, and extracellular matrix organization), while the decreased-G2 are largely associated with pluripotent state (Fig. 3i, j). Similar analysis performed for genes located closest to Sall4$^{K5A}$ specific CUT&Tag peaks showed that Sall4$^{K5A}$-bound genes largely overlap with group 3 and 6 (Supplementary Fig. 4e). Gene ontology analysis of the overlap genes revealed that Sall4$^{K5A}$-G3 and Sall4$^{K5A}$-G6 are associated with somatic features such as extracellular matrix organization and cell adhesion (Supplementary Fig. 4f). Together, these results suggest that Sall4 plays a crucial role in the transcriptional regulation of various important biological processes during the process of reprogramming.

NuRD complex cooperates with transcription factors to regulate gene expression at chromatin level by ATP-dependent chromatin remodeling and histone deacetylase activities. The modification of histone tails such as H3K27ac is tightly coupled to chromatin accessibility and gene expression. Thus, dissociation between NuRD complex and Sall4$^{K5A}$ can result in failure of NuRD-mediated deacetylation of histone H3K27 at somatic gene loci. We then ask whether the high levels of H3K27ac in Sall4$^{K5A}$ are due to the lack of NuRD subunits. To this end, we performed CUT&Tag experiments with anti-Gatad2b antibody during Sall4$^{WT}$ and Sall4$^{K5A}$ reprograming (Supplementary Fig. 4g). Firstly, we analyzed genome-wide occupancy of Sall4 CUT&Tag dataset in Sall4$^{WT}$ and Sall4$^{K5A}$, generating three groups, WT and K5A co-binding regions (common), K5A specifically binding (S$^{K5A}$ speci) and WT specifically binding (S$^{WT}$ speci) (Fig. 3k). Then each group was subdivided into three subgroups by GATAD2B CUT&Tag dataset in WT or K5A condition (Fig. 3k). Among genomic regions occupied by both Sall4$^{WT}$ and Sall4$^{K5A}$, large numbers of Gatad2b-bound peaks (7199)

were identified only in Sall4$^{WT}$ reprogramming, which were not bound by Gatad2b in Sall4$^{K5A}$. Similar Gatad2b binding pattern was also found among Sall4$^{WT}$ specific regions (463) (Fig. 3l). The weak signals at sites occupied by Gatad2b in Sall4$^{K5A}$ condition reveals that Gatad2b genomic binding ability is partly compromised when Sall4 dissociates with NuRD complex. Again, those loci are highly enriched in TF motifs from FRA1, ATF3, JUNB, BATF, JUN, that are all AP1 TFs. These results suggest that disruption of the Sall4-NuRD axis leads to mislocalization of NuRD and failure to close somatic loci enriched with AP1 TFs (Supplementary Fig. 4h).

To further identify candidates, we compared annotated nearest genes within each cluster and found overlaps between annotated nearest genes and group 3 and 6 (Supplementary Fig. 4i, j). Gene ontology analysis of the overlap genes in Gatad2b lost-G3 and Gatad2b lost-G6 are associated with angiogenesis, transmembrane transport, respectively (Supplementary Fig. 4k). To further validate our predictive analysis and narrow down the scale of candidates, we found 16 overlap genes by selecting the common genes from group1 of shGatad2b RNA-seq and Gatad2b lost-G3 (Supplementary Fig. 4l). Those genes are rapidly downregulated after JGES induction but remain higher expression with knocking down of Gatad2b or Sall4$^{K5A}$. When we overexpressed 15 of the candidates during JGES reprogramming, 11 genes of them lead to significantly decreased reprogramming efficiency (Supplementary Fig. 4m), highlighting the requirement of NuRD complex for inactivation of somatic program during JGES reprogramming.

Next, we ask whether above sites existed in open chromatin conformation with active histone marks and ATAC-seq signal, we compared loci with elevated H3K27ac (9828) with those lost Gatad2b binding (7199 + 463) to identified a total of 610 common loci and found a reciprocal relationship between Gatad2b and H3K27ac (Fig. 3m, green vs red). Motif enrichment analysis demonstrate that those loci are mostly bound by TFs such as AP-1, Atf3, JunB and BATF (Supplementary Fig. 4n). Further analyzing chromatin accessibility dynamic shows that the 610 loci were accessible in MEFs and became more accessible by day0 and day1, then loss of accessibility progressively and became inaccessible in ESCs. Of note, an increase in ATAC-seq signal was observed in Sall4$^{K5A}$ on day1 (Fig. 3n). These results suggest that NuRD complex was involved in closing of somatic chromatin. To determine gene expression in the 610 regions, we performed an integrative analysis of CUT&Tag and RNA-seq data. We then focused on genes that fail to be downregulated in G3 and abnormally activated in G6 during Sall4$^{K5A}$ reprogramming (Fig. 2i). There were 39 genes in G3 and 13 genes in G6 accompanied with elevated H3K27ac and Gatad2b lost (Supplementary Fig. 4o). Interestingly, *Tgfb3, Runx1, Chd3* and *Rasa3* were identified with a slower inactivation pattern in Sall4$^{K5A}$ than Sall4$^{WT}$ (Supplementary Fig. 4p). Based on the results that slow inactivation of somatic genes such as *Rasa3, Htra1, Fzd2* either in shGatad2b or Sall4$^{K5A}$, we propose that this may represent a new class of barrier genes for cellular reprogramming. Indeed, we show that knocking down Rasa3 facilitates and its overexpression inhibits JGES reprogramming, respectively (Supplementary Fig. 4q, r). More importantly, knocking down Rasa3 appears to rescue the defect caused by Gatad2b knockdown or mutated Sall4 to some extent (Supplementary Fig. 4s, t). Together, these results suggest that the NuRD complex is required to close somatic chromatin loci that encode barriers during JGES reprogramming.

## Rescue of K5A Sall4 mutant by grafting N12 onto Jdp2

JGES may convert MEFs to iPSCs by closing somatic chromatin and opening pluripotent ones, similar to OSKM and chemical reprogramming[53,59]. Given the essential role demonstrated here for the collaborative interactions between Sall4 and NuRD complex directing somatic program inactivation, we may rescue Sall4 mutant(s) by reconstituting an alternative chromatin closing pathway. We first

analyzed the genome-wide occupancy of Jdp2, Glis1, Esrrb, Sall4, and Gatad2b by CUT&Tag and show that they bind to diverse loci (Supplementary Fig. 5a), suggesting that there may be a division of labor among chromatin binding proteins, with Jdp2 binding to somatic chromatin and Esrrb binding to pluripotent loci. We then designed and constructed synthetic factors by fusing N12 (the NuRD interaction motif) to N terminal of Jdp2 (native Jdp2), Esrrb and Glis1, and introduced these factors with Sall4$^{K5A}$ during reprogramming to show that Jdp2$^{N12}$ (synthetic Jdp2), not Glis1 $^{N12}$ nor Esrrb $^{N12}$, can rescue Sall4$^{K5A}$ (Fig.4a, b) and Sall4$^{delN12}$ defect (Supplementary Fig. 5b, c).

Then we performed RNA-seq on JGES, JGES$^{K5A}$, J$^{N12}$GES$^{K5A}$ and JGE. PCA indicated that JGE and JGES$^{K5A}$ differ significantly from JGES at gene expression levels. In contrast, J$^{N12}$GES$^{K5A}$ share similar dynamic with JGES reprograming despite a delay (Supplementary Fig. 5d). To gain mechanistic insight into why synthetic Jdp2 has this unique ability, we performed IP-MS with JDP2 antibody, and show that it can recruit subunits of NuRD complex (Fig. 4c, processed data provided in supplementary data1). Next, to explore the function of N12 during reprogramming, MEFs were infected with Esrrb, Glis1, Sall4$^{K5A}$, and synthetic or native Jdp2.Then we performed JDP2 CUT&Tag to map their genomic occupancy. A total of 110402 peaks from both samples including 57104 common peaks, 23452 JDP2$^{N12}$ specific binding peaks and 29846 JDP2$^{WT}$ specific binding peaks was identified (Fig. 4d). We show using de novo motif calling that both synthetic and native JDP2 binds to genomic regions enriched for AP-1 family motifs such as Fra1/2, Atf3, JunB and Fosl2 (Fig. 4e). We further performed ATAC-seq to compare the chromatin accessibility of reprogramming mediated by synthetic and native Jdp2 (Fig. 4f). We classified ATAC-seq peaks into native-specific (41264), synthetic-specific (6611) and common loci (49890) (Fig. 4g), and show that common loci are enriched in motifs for somatic state TFs such as Fra1/2, Atf3, JunB, and Fosl2 (Fig. 4h). Importantly, we identified peaks that are closed in synthetic Jdp2 but remain open in native Jdp2 during reprogramming to show that transcript level correlates well with chromatin accessibility for genes such as *Tgfbr2*, *Htra1*, *Bmp1*, *Nrp2*, *Wnt5a*, *Col6a3*, *Igfbp7*, and *Bmp4* (Fig. 4h, Supplementary Fig. 5e), all shown to inhibit JGES reprogramming (Supplementary Fig. 4m). Together, these data indicate that N12 is required for Jdp2 to close somatic state related chromatin.

To validate Jdp2$^{N12}$ may recruit NuRD complex to orchestrate chromatin remodeling and trigger somatic program inactivation. we performed GATAD2B CUT&Tag experiment during Jdp2$^{N12}$ and Jdp2$^{WT}$ reprogramming in combination with Esrrb, Glis1, and Sall4$^{K5A}$ on day1. First, we categorized the JDP2$^{N12}$ and JDP2$^{WT}$ CUT&Tag peaks into the simplest tier of JDP2-Common, JDP2$^{N12}$ specific, and JDP2$^{WT}$ specific (Fig. 4d), and then we analyzed GATAD2B binding density for above three regions during Jdp2$^{WT}$ and Jdp2$^{N12}$ reprogramming (Fig. 4i, Supplementary Fig. 5f, g). For JDP2$^{N12}$-specific regions, we observed higher GATAD2B binding density in Jdp2$^{N12}$ reprogramming. Conversely, the JDP2$^{WT}$-specific regions exhibited increased GATAD2B occupancy in Jdp2$^{WT}$ reprogramming. Collectively, these results suggest that grafting the NuRD interacting motif onto Jdp2 to engage somatic specific regions can functionally rescue the disrupted Sall4-NuRD axis (Fig. 4j).

## Discussion

We show here that JGES reprogramming differs markedly from the classic Yamanaka factors OSKM in that JGES relies on Sall4 to engage endogenous NuRD to close open chromatin in MEFs while OSKM relies on pioneering factors to open chromatin in MEFs[10,60]. This Sall4-NuRD axis may play similar roles in normal development and disease processes.

While sharing the same starting cells, i.e., MEFs, and final outcome, i.e., iPSCs capable of chimera formation and germline transmission, JGES may orchestrate reprogramming quite differently from OSKM, thus, offering a rare opportunity to compare and contrast their strategies in mediating cell fate decisions. The pioneering model of reprogramming has been proposed for OSKM based on their ability to bind to chromatin not normally accessible by transcription factors[60–62]. It has been thought that this pioneering function is a critical feature of OSKM reprogramming, these factors are not normally expressed in MEFs, thus, their binding motifs are buried mostly in chromatin in closed forms. However, the pioneering model does not provide explanations on how the open chromatin in MEFs are closed. Our earlier work suggests that OSK activates endogenous factors such as Sap30 that will engage Sin3A to close open chromatin in MEFs during early phase of reprogramming[53]. By identifying the Sall4-NuRD axis here, we propose that similar chromatin closing event should be the initiating event in cell fate transition.

Our findings appear to contradict multiple earlier studies that implicated NuRD subunits as a negative rheostat in reprogramming, including one that found that Gatad2a and Chd4 depletion resulted in up to 100% iPSC derivation efficiency[28]. Our study is different from these other studies in a number of ways, including the reprogramming cocktail we used and the reprogramming conditions. Mor and colleagues used knockdown experiments by siRNA or knockout to inhibit Gatad2 or Chd4 during reprogramming, in contrast to our retrovirus delivery[34]. While in our reprogramming system, the MEFs were infected with JGES retrovirus for reprogramming, they opted for transgenic "secondary reprogramming" embryonic fibroblasts (MEFs) that carry TetO-inducible OKSM for iPS induction. Furthermore, Mor and colleagues discovered that repressing Mbd3 and Chd4 with targeted siRNA prior to OKSM induction hampered the reprogramming process. Meanwhile, Santos and colleagues reported a positive role for MBD3/NuRD in transcription factor-mediated reprogramming of neural stem cells and epiblast stem cells to naive stem cells, implying a context-dependent role for the NuRD complex in pluripotency induction[32]. Besides, we have reported that MEFs induced with OKSM and 7 F (Jdp2, Esrrb, Sall4, Nanog, Kdm2b, Mkk6, Gkis1) follow distinct molecular trajectory during 7-day course to arrive final naïve state[14]. Mor and colleagues proposed a model that Gatad2a/Mbd3 represses the same genes that OKSM try to reactivate. However, Our JGES reprogramming system has shown a preference for collaboration with the NuRD complex to effectively deactivate somatic cell-specific genes during the early stages of reprogramming.

Sall4-NuRD interaction has been reported in cancer and in development processes such as hematopoiesis and neurogenesis[63–66]. However, the precise mechanism of its role in carcinogenesis has not been fully understood. In light of our finding here, one may argue that Sall4 functions to silence critical cell fate regulators through NuRD and then promote cell fate towards cancerous direction (Fig. 4j). If so, our work may lead to better models for therapeutical development.

## Methods

### Mice

OG2 transgenic mouse (CBA/CaJ x C57BL/6 J) were purchased from the Jackson laboratories (Mouse strain datasheet: 004654). Animals were individually housed under a 12 h light/dark cycle and provided with food and water *ad libitum*. Our studies followed the guidelines for the Care and Use of Laboratory Animals of the National Institutes of Health, and the protocols were approved by the Committee on the Ethics of Animal Experiments at the Guangzhou Institutes of Biomedicine and Health.

### DNA constructs, cell lines, and cell culture

All constructs for in vitro expression were cloned to pMXs plasmids, and shRNAs were cloned to pSuper plasmids. MEFs were isolated from E13.5 mouse embryos regardless of sex from crossing male Oct4-GFP transgenic allele-carrying mice (CBA/CaJ 3 C57BL/6 J) to 129S4/SvJaeJ female mice around 6–8 weeks old. Briefly, the integral organs, the tail, the limbs and head were removed. The remaining tissues were cut into small pieces and then dissociated by digestive solution (0.25% trypsin:

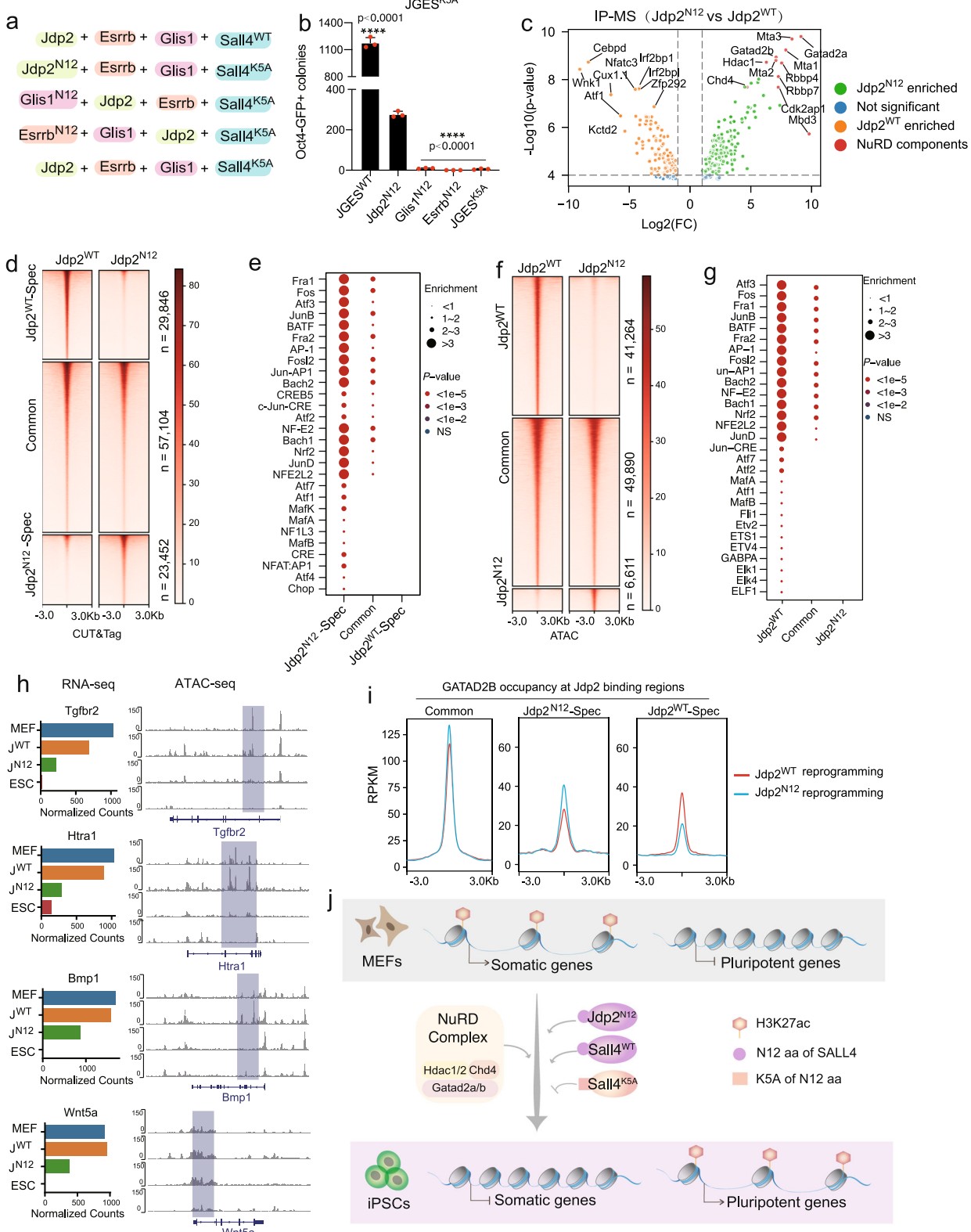

0.05% trypsin =1:1; GIBCO) for 15 min at 37°C to obtain a single cell suspension. The isolated MEFs were seed onto 0.1% gelatin-coated culture dish DMEM-high glucose (Hyclone) contain 10% FBS (GIBCO), 1% GlutMAX (GIBCO),1% sodium pyruvate (GIBCO) and 1% NEAA (GIBCO), which was defined as fibroblast medium. Plat-E cells were maintained in DMEM high-glucose media (Hyclone) supplemented with 10% FBS (NTC, SFBE, HK-026). Mouse ESCs derived from embryos

at home from Oct4-GFP transgenic mice and iPSCs (Male or female) derived in the study were cultured feeders-free with N2B27-2i medium (50% (v/v): high-glucose DMEM (Hyclone), 50% (v/v) knock out DMEM (GIBCO), N2 (GIBCO), B27 (GIBCO), 1% sodium pyruvate (GIBCO), 1% non-essential amino acids (GIBCO), 1% GlutaMAX (GIBCO), 0.1 mM 2-mercaptoethanol (GIBCO),1000 U/ml leukemia inhibitory factor (LIF) (Millipore), and the 2i inhibitors, 3 mM CHIR99021 (Sigma), and

**Fig. 4 | Rescue of K5A Sall4 mutant by grafting N12 onto Jdp2. a** N-terminal 12 AAs of SALL4 was fused individually to the other three factor in JGES system, JDP2, GLIS1, ESRRB, which was named as followed, Jdp2$^{N12}$, Glis1$^{N12}$, Esrrb$^{N12}$. **b** Bar plot of numbers for Oct4 GFP positive iPS colonies induced by Jdp2$^{N12}$: (Jdp2$^{N12}$+Glis1+Esrrb+ Sall4$^{K5A}$), Glis1$^{N12}$:(Jdp2+Glis1$^{N12}$+Esrrb+Sall4$^{K5A}$), Esrrb$^{N12}$:(Jdp2+Glis1+Esrrb$^{N12}$+Sall4$^{K5A}$). Data are mean ± s.d., two-sided, unpaired t test; $n = 3$ independent experiments, ***$p < 0.001$, ****$p < 0.0001$. Source data is provided as a Source Data file. **c** Volcano plots of enriched proteins by Jdp2$^{N12}$ and Jdp2$^{WT}$ pull down followed by MS analysis. NuRD complex subunits were dots in brownish-red. Jdp2$^{WT}$ enriched protein were dots in orange. Jdp2$^{N12}$ enriched protein were dots in green. IP-MS experiments were performed in triplicates and a two-sided $t$ test was applied. $P = 0.00001$ and fold change = 1.5 were used as threshold. **d** Heat map showing the density of Jdp2 CUT&Tag reads by comparing the sample on day1 of the reprogramming by Jdp2$^{N12}$ and Jdp2$^{WT}$ in combination with GES$^{K5A}$. **e** Enriched motifs of the regions in Fig. 4d. **f** Heat map showing the density of ATAC-seq reads by comparing the sample on day1 of the reprogramming by Jdp2$^{N12}$ and Jdp2$^{WT}$ in combination with GES$^{K5A}$. **g** Enriched motifs of the regions in Fig. 4f. **h** selected genomic views of ATAC-seq data are shown (right) for the indicated genes in Jdp2$^{N12/WT}$ and Sall4$^{K5A}$ reprogramming on day 1, MEF, ESC with corresponding RNA expression (left). **i** Pileup of GATAD2B CUT&Tag signal for regions in (Fig. 4d) during Jdp2$^{WT}$ and Jdp2$^{N12}$ reprogramming. Jdp2$^{N12}$ and Jdp2$^{WT}$ are indicated by blue and red, respectively. (Upstream 3 kb and downstream 3 kb of the peak center). **j** A model to illustrate the cooperation between Sall4 with NuRD during iPSC induction in JGES reprogramming system.

1 mM PD0325901 (Sigma) or mES medium for karyotype analysis: high-glucose DMEM (Hyclone),15%(v/v)FBS(GIBCO), 1% sodium pyruvate (GIBCO), 1% non-essential amino acids (GIBCO), 1% GlutaMAX (GIBCO), 0.1 mM 2-mercaptoethanol (GIBCO),1000 U/ml leukemia inhibitory factor (LIF) (Millipore),.All the cell lines have been confirmed as mycoplasma contamination free with the Kit from Lonza (LT07-318).

## iPSCs generation

This protocol started with production of the retro-virus. Plat-E cells were seeded at the concentration of $7.5 \times 10^{6}$–$8.5 \times 10^{6}$ Cells per 10 cm dish uniformly and then were cultured in high-glucose DMEM (HyClone, SH30022.01) supplemented with 10% FBS (NTC, SFBE, HK-026) medium (10% FBS) for 12–16 h to reach a 70–80% confluent. The next step is plasmid transfection. For each 10 cm dish, Replacement of the Plat-E cells medium with 7.5 mL fresh 10% FBS should be applied firstly. A modified calcium phosphate transfection method was conducted as follows: each plasmid should be manufactured in an individual tube, 1068 μL ddH$_2$O, 25 mg plasmid, 156.25 μL 2 M CaCl2, 1.25 ml 2×HBS were added in order to a total volume of 2.5 ml, mix the liquid immediately after adding 2×HBS, after incubate for 5 min at room temperature, the mixture should be gently transferred into the Plat-E cell. Replace the medium with 10 ml 10% FBS within 10–16 h after transfection. And then, the retrovirus should be collected twice, 48 h and 72 h after transfection, the supernatant containing the virus was collected at each time by a syringe and filter through a 4.5 mm filter and a 10 ml fresh 10% FBS medium was added to the Plat-E cell dish after the first collection, the virus could be stored at room temperature for 48 h at most. Thawing the frozen Passage 1 OG2 MEF (mouse embryonic fibroblast) into a 6 cm dish with 10% FBS medium and cultured in a 5% CO$_2$ incubator while conduct the transfection. Then split the MEFs to P24 plate at $1.5 \times 10^{4}$ cell density per well before infected with virus when reach a 100% confluence. MEF cells should be also infected for twice. Mix the virus stock at proper volume and one volume fresh 10% FBS medium, then mix polybrene with the mixture to a final concentration of 4 mg/ml before infection. The second virus infection was conducted 24 h later. At post-infection Day0, replace the virus contained medium with fresh reprogramming medium iCD3 or chemical screening medium. Change medium every 24 h and observe the morphology change. GFP$^+$ colony will appear at day 2 to day 3, GFP$^+$ clones are photoed by living cells station (NIKON, Bio Station CT) and counted by Image-J using particles analysis.

## ICD3 establishment

The chemical screening library include 630 chemicals is consist of 10 signal pathways relative to Tyrosine Kinase/Adaptors ($n = 142$), PI3K/Akt/mTOR Signaling ($n = 96$), Chromatin/Epigenetic ($n = 86$), Immunology/Inflammation ($n = 59$), JAK/STAT Signaling ($n = 51$), MAPK Signaling ($n = 54$), Angiogenesis ($n = 37$), Stem Cell ($n = 27$), Metabolism ($n = 16$), Neuroscience ($n = 14$), and others ($n = 48$).Before the chemical screening, by deleting LiCl which represses 7 F reprogramming efficiency from iCD1 we developed iCD2, and then, we deliver the chemicals one by one into iCD2 at 1 μM and 5 μM concentration, after JGES virus transfection, cells were treated with iCD2 plus chemicals for 7 days and GFP$^+$ clones were measured then. TOP5 chemicals were then combined with each other for next round reprograming, at the end of the screening, we established iCD3 reprograming culture medium.

## Immunofluorescence

Cells growing on glass slide (NEST, 801007) were washed 3 times with PBS, then fixed with 4% PFA for 0.5 h, after washing 3 times in PBS and subsequently penetrated and blocked with 0.1% Triton X-100 and 3% BSA for 0.5 h at room temperature. Then, the cells were washed 3 times and incubated with primary antibody diluted with 3% BSA for two hours at room temperature or over-night at 4 degrees. After 3 washes in PBS, the cells were incubated for one hour in second antibodies diluted with 3% BSA. After washing 3 times in PBS cells were then incubated in DAPI diluted with PBS for 2 min. Then, the glass slide was mounted on the slides for observation on the confocal microscope (Zeiss 710 NLO). The following antibodies were used in this project: anti-Flag (Sigma Aldrich, F1804 1:200)

## Co-immunoprecipitation and western blot

To perform co-immunoprecipitation, Cells were digested with 0.25% trypsin and washed for 3 times in PBS, whole cell extracts were prepared using lysis buffer (50 mM Tris pH 7.4, 200 mM NaCl, 10% Glycerol, 1% NP40,1 mM EDTA) with fresh added 1x Complete Protease inhibitors (Sigma, 1187358001) and 1% PMSF, incubated for 15 min on ice and then 1 h at 4 °C on a rotation wheel. Soluble cell lysates were collected after maximum speed centrifugation at 4 °C for 15 min, the supernatant was incubated with anti-FLAG beads, DYKDDDDK (Themo Fisher, A36797) overnight at 4 °C on a rotation wheel. Beads were then washed three times with cell wash buffer (50 mM Tris pH 7.4, 200 mM NaCl, 10% Glycerol, 0.01% NP40,1 mM EDTA) for 5 times. After completely removal of cell wash buffer, immunoprecipitated proteins with FLAG beads were boiled at 100-degree water in loading buffer (4% SDS,10% 2-Mercaptoethanol, 20% Glycerol,0.004% Bromophenol blue, 0.125 M Tris Ph 6.8) for 10 min, Whole protein extract were stored at −80 degree and avoid freeze and thaw cycle. To perform western blot, Total proteins or IP extract were analyzed by SDS-PAGE and then transferred to PVDF membrane (Millipore). After incubated with indicated antibodies, the membrane was exposed to X film. NuRD Complex Antibody Sampler Kit (CST,8349 T,1:1000), Anti GATAD2B (Abcam, ab224391,1:1000), anti RBBP4(Novusbio NB500-123,1:1000), anti-FLAG (Sigma Aldrich, F1804 1:1000), anti SALL4 (Abcam, ab29112,1:1000), anti-H3K27ac (Abcam ab4729,1:1000), were used.

## Mass spectrometry analysis

Peptides after digestion were separated by AcclaimTM PepMapTM 100 C18 column (Thermo, 164941) using a 140 min of total data collection (100 min of 2–22%, 20 min 22–28% and 12 min of 28–36% gradient of B buffer (which containing 80% acetonitrile and 0.1% formic acid in H2O) for peptide separation, following with two steps washes: 2 min of 36–100% and 6 min of 100% B buffer) with an Easy-nLC 1200 connected online to a Fusion Lumos mass spectrometer (Thermo). Scans

were collected in data-dependent top-speed mode with dynamic exclusion at 90 s. MaxQuant version 1.6.0.1 search against Mouse Fasta database was used to analyze raw data, with label free quantification and match between runs functions enabled. DEP package was used to analyze and visualize the output protein group.

### RNA-seq and data analysis
Total RNA was isolated with TRIzol (Invitrogen). Libraries were prepared using the VAHTS mRNA-seq v2 Library Prep Kit for Illumina (Vazyme, NR601-01/02,) with 1 μg RNA per sample following the manufacturer's instruction. Sequencing was performed using an illumina nova seq instrument at GUANGZHOU IGE BIOTECHNOLOGY LTD, (Gunagzhou, China). To analyze the gene expression, reads were aligned to the reference transcriptome using RSEM[67] (v. 1.2.28) and the index built by RSEM with the mouse genome, mm10, and annotated to gene by annotables (v. 0.1.90) in R. Then, DESeq2[68] (v. 1.26.0) was used for data normalization and differential expression analysis. Differentially expressed genes were defined by Wald test (Benjamini-Hochberg-corrected $P$-value < 0.05 and absolute fold change > = 1.5) and Likelihood ratio test (Benjamini-Hochberg-corrected $P$-value < 0.05) for time course experiments. Gene ontology analysis was performed using clusterProfiler[69] (v. 3.14.3)

### ATAC-seq and data analysis
ATAC-seq library construction was performed using TruePrep DNA Library Prep Kit V2 for Illumina (Vazyme, TD501-01) and TruePrep Index Kit V2 for Illumina (Vazyme, TD202). Around 50,000 living cells were collected for each sample and the ATAC library was sequenced on a illumina nova6000 and carried out by Berry Genomics Corporation, (Beijing,China). All of the sequencing data were aligned to the mouse genome assembly (mm10) using bowtie2[70] (v. 2.3.5.1) with the following options: -p 20 –very-sensitive -k 10. Then, sambamba[71] (v. 0.6.6) was used to sort and remove duplicate reads with the following options: [XS] == null and not unmapped and not duplicate. Alignment BAM files were transformed into read coverage files (bigWig format) using deepTools[72] (v. 3.5.1) using the RPKM normalization method. peaks were called using genrich (v. 0.6) with options: -j -y -r -m 30 -e MT -v -q 0.01. Then, peaks from different sample were merged to a peak set by DiffBind[73] (v. 2.14.0) using RPKM or read count. Differential binding region was defined based on the peak set with read count by DESeq2 using Wald test (Benjamini-Hochberg-corrected $P$-value < 0.05 and absolute fold change > = 1.5) and Likelihood ratio test (Benjamini-Hochberg-corrected $P$-value < 0.05) for time course experiments. Open-close state change was based on peak set of RPKM with boundary of 2. Motif analysis was performed using HOMER[74] (v.4.11). Peak was annotated to gene loci by ChIPseeker [69](v. 3.20.1).

### CUT&Tag and data analysis
CUT&Tag library construction was performed using Hyperactive In-Situ ChIP Library Prep Kit for Illumina (pG-Tn5) (Vazyme, TD901) and TruePrep Index Kit V2 for Illumina (Vazyme, TD202). Around 100,000 living cells were collected for each sample and the CUT&Tag library was sequenced on a illumina nova6000 and carried out by Berry Genomics Corporation, (Beijing,China).

Sequenced reads were aligned to the mouse reference genome (mm10) using Bowtie2 with the parameters:–end-to-end–very-sensitive–no-unal–no-mixed–no-discordant–phred33 -I 10 -X 700. Then, sambamba was used to sort and remove duplicate reads with the following options: [XS] == null and not unmapped and not duplicate. Peaks were called using MACS2[75] (v. 2.2.6) with the default parameters. Reads that mapped to mitochondrial DNA or unassigned sequences were discarded. For paired-end sequencing data, only concordantly aligned pairs are retained. Alignment BAM files were transformed into read coverage files (bigWig format) using deepTools using the RPKM normalization method. For the sample without repeat,

differential binding region was analysis by manorm[76] (v. 1.3.0) with $P$-value < = 0.01. For the sample with repeats, differential binding region was analysis by DESeq2 based on the peak set produced by DiffBind, which is the same as ATAC-seq analysis. Motif analysis was performed using HOMER. Peak was annotated to gene loci by ChIPseeker.

### Karyotype Analysis
Karyotype analysis was performed according to protocol published previously[8,77–79]. Briefly, $5 \times 10^5$ cells were seeded on 10 cm cell culture dishes and incubated for 48 h to reaching a 90% confluence. The cells were treated with fresh medium containing 0.2 μg/mL colchicine incubating for 2 h. The treated cells were collected and resuspended in 7 mL of 37 °C KCl hypotonic solution. After hypotonic treatment, nuclei were collected by centrifugation. Using a freshly configured Carnot fixation solution, pre-immobilize the nucleus for 3 min and then sample was collected and fixed again at 37 °C for 40 min. After fixation, the nuclei were collected and then resuspend. Clean slides were soaked cold water before use. Draw up a few drops of resuspended cells onto chilled and clean slides and spread them and then in a 75 °C oven for 3 h. Add trypsin to the staining vat and preheated in 37 °C. The dry slides were digested by trypsin for 8–12 s then terminated by saline. After that, the slides were stained with filtered Giemsa for 4 min then washed by PBS and ddH2O. Check the chromosomes with the microscope. Count at least 20 cells. Significant problem if more than 4 cells have more or less than 40 chromosomes (mouse).

### Statistics and reproducibility
Data are presented as mean ± s.d. as indicated in the figure legends. Unpaired two-tailed student t-test, The $P$-value was calculated with the Prism 6 software. A $P < 0.05$ was considered as statistically, *$P < 0.05$, **$P < 0.01$, ***$P < 0.001$, ****$P < 0.0001$. No statistical method was used to predetermine sample size. All experiments were replicated at least three times, and data are shown as means with SEM. No specific randomization or blinding protocols were used.

### Reporting summary
Further information on research design is available in the Nature Portfolio Reporting Summary linked to this article.

## Data availability
The data supporting the conclusions of this study, including CUT&Tag for H3K27ac, Sall4, Jdp2, Gatad2b, Esrrb and Glis1 are available at GEO under accession GSE199612. The ATAC-seq and RNA-seq data were from GSE199609 and GSE199613. The RNA-seq data of MEF and ES cells was obtained from GSE127927. The mass spectrometry proteomics data have been deposited to the ProteomeXchange Consortium (http://proteomecentral.proteomexchange.org) via the iProX partner repository with the dataset identifier PXD041704. Source Data for Figs. 1a, d, 2b, 4c, and Supplementary Figs 1b, e, g, h, 2a, 3d, e, g, 4m, q, r, s, t, 5c are provided with the manuscript. The authors declare that all data supporting the findings of this study are available within the article and its supplementary information files or from the corresponding author upon reasonable request. Source data are provided with this paper.

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

## Acknowledgements

This work was supported in part by the National Natural Science Foundation of China (31830060, 92068201, 32000502, 81974019), Major State Basic Research Development Program (2017YFA0504100), Guangdong Science and Technology Project (2020B1212060052). The Research Team Project of Natural Science Foundation of Guangdong Province of China (2017A030312007), Natural Science Foundation of Guangdong Province of China (2019A1515012032), National Key Research and Development Program of China (2018YFA0108700, 2017YFA0105602), NSFC Projects of International Cooperation and Exchanges (81720108004). Guangdong Provincial Special Support Program for Prominent Talents (2021JC06Y656), Science and Technology Planning Project of Guangdong Province (2022B1212010010), Guangzhou Science and Technology Plan Project (202201000006). The Special Project of Dengfeng Program of Guangdong Provincial People's Hospital (DFJH201812; KJ012019119; KJ012019423), High-level Hospital Construction Project (DFJHBF202110).

## Author contributions

B.W. and J.M. performed the main experiments; C.L., L.L., H.F., C.Z., and X.H. performed the bioinformatic analysis; J.M., S.F., and Y.H. performed the cell culture experiments; L.W. performed the chemical screening experiment; J.K., H.L., and X.Y. performed the RNA-seq experiment; J.G. construct the plasmids; L.G. performed and X.Z. analysis the IP-MS experiment; P.Z., J.C., and J.L. supervised the bioinformatics analysis; D.P. supervised and conceived the whole study, wrote the manuscript, and approved the final version.

## Competing interests

The authors declare no competing interests.
