## [Peer Review File · Nature Communications]

The NuRD complex cooperates with SALL4 to orchestrate reprogrammingREVIEWER COMMENTS

Reviewer #1 (Remarks to the Author):

The authors identify Sall4 as being a particularly important component in their alternative remodelling set of transcription factors so they ask what its molecular interactors are by performing an IP-Mass Spec experiment. They identify a well-known Sall4 interactor, NuRD, and then characterise what that might be doing. They perform extensive RNAseq, ChIP-seq and ATAC-seq on reprogramming cells, but rather than use this to determine an order of events for the reprogramming process and the role of Sall4 in it, the data are mined to find evidence to support their model, which is that Sall4 recruits NuRD to silence things. In the end they identify 16 genes at which all of the data indicate this is exactly what is happening. So while what they say appears to be true for these loci, there are hundreds to thousands of other loci for which their evidence does not support the model. Is everything else that Sall4 does really irrelevant?

An alternate approach would be to identify any previously unknown interactors and ask whether they contribute to Sall4 function in reprogramming. The authors' data show that Sall4 likely has further functions outside of the well-trodden path of NuRD interaction: there are considerably more iPS colonies formed when knocking down Gatad2b or Chd4 than if Sall4 is left out of the reprogramming mix. Even the K5A mutation produces more colonies than if Sall4 is left out. Does this mean that Sall4 also has a NuRD-independent function? If so, what is it? Perhaps the proteomics data can provide some clue?

Specific comments:

The author suggest in the Abstract that "Selective closing of open chromatin may be an early step in cell fate decision." This does not seem like a new idea to me. The first time I can recall this being highlighted for reprogramming was in the following publication, but I suspect there have been many other papers as well:

Chronis C, Fiziev P, Papp B, et al. Cooperative Binding of Transcription Factors Orchestrates Reprogramming. *Cell*. 2017 Jan;168(3):442-459.e20. DOI: 10.1016/j.cell.2016.12.016. PMID: 28111071.

Figure 1c shows that they have identified a large number of interactors. Prominent amongst them are NuRD components. Sall4 and NuRD are well known to co-purify in a range of cell types, so this result is expected, but it is reported in the text as if this were a novel finding.

Figure 1c shows a large number of interactors, but only two which are not NuRD components (Parp, Lig3) are labelled. There is no indication of why these two proteins are also labelled, unless they have been mistaken for NuRD components? What are all of these other interactors? How many also purify with the K5A mutant? These proteomics data are not made available, nor have the authors indicated that they have deposited these data in any repositories. There is no mention in the Methods of how these experiments were performed.

There is no description of how the knockdowns were done or what the timelines are. Knocking down/out Chd4 and Gatad2a/b has been reported to result in cells undergoing cell cycle arrest: do the authors know that failure to reprogramme is not simply caused by the cells exiting the cell cycle?

Figure 1d, extended fig 2: The authors are stating the knocking down Chd4 and Gatad2 proteins has the most pronounced effect on reprogramming. The point is to be able to compare the effects for different components. so the data in these two figures should be combined and shown in one main fig: (Extended fig 2a has a typo in the figure: "Kncok NuRD Subunits")

Figure 1e: very difficult to see the data. Appears to be two replicates, but I can't see the final Chd4 KD datapoint. It looks like WT and Chd4 kd are about the same at D7...what do we conclude from this? How were the arrows drawn in?

Line 110: the authors conclude that their data "suggested that Sall4 recruit NuRD complex to silence the somatic gene." So far they show two perturbations which inhibit reprogramming: leaving Sall4 out of the mix, and knocking down NuRD components. In both cases there will be a failure to progress down the reprogramming trajectory in terms of gene expression, chromatin changes, etc. This does not mean that the two things are necessarily working together, just that they both fail to reprogramme.

The data with the K5A mutation argues very strongly that the majority of the effect they are showing relies on a Sall4-NuRD interaction. They show through mutational analysis that most of the zinc fingers also play a contributing role to Sall4 function in reprogramming (assuming none of these impact protein stability), but this is not explored further.

Lines 151-156: they specify that K5A cells resemble shGatad2b cells. Does this imply that they don't resemble shChd4 cells? Any idea why not?

Line 169: 14.5% of gene loci show an increase in K27Ac in the K5A cells vs WT and they focus on these. Yet there is no justification given for why they ignore the very similar number of loci showing a decrease in K27Ac, other than the former examples fit their model.

Lines 174-175: the authors conclude that Sall4 normally recruits NuRD to K27Ac-high regions to close them. Yet they only do K27Ac ChIP for one time point: Day1. So we don't know that these sites show any change in K27Ac normally across the time course. How do they behave across the ATAC-seq time course? Do they get closed?

Lines 176-177: "NuRD contains histone deacetylases capable of removing H3K27ac, resulting in nucleosome compaction and gene silencing" NuRD-mediated nucleosome compaction and gene silencing has been shown to precede H3K27 deacetylation in two different studies. I am not aware of any studies showing that H3K27Ac results in compaction and gene silencing.

Lines 198-199: This is quite a small number of genes. When comparing datasets like this there will always be overlaps, so how do we know this really is significant? Perhaps a more relevant question is why don't we see a bigger effect if Sall4 is recruiting NuRD to repress genes? If this is really an important mechanism then why can they only find 16 genes that support their model?

Line 257: They state that the Sall4-NuRD connection has been reported in cancer. While true it's also been reported in many other cell types.

Reference 27 is incomplete.

Reviewer #2 (Remarks to the Author):

The somatic cell reprogramming process is suitable for studying cell fate control. In this study, the authors demonstrated that mouse embryonic fibroblasts (MEFs) could be reprogrammed with high efficiency by introducing *Jdp2*, *Glis1*, *Esrrb* and *Sall4* (JGES) and culturing in a specific chemically defined medium (iCD3) and that the *Sall4*-NuRD interaction is essential for the reprogramming with JGES. Moreover, they want to show that *Sall4* recruits the NuRD complex to the open chromatin regions, including many somatic genes, leading to closing the chromatin regions to enhance the

initiation event in the reprogramming.

This study is potentially interesting because it might provide us with new insight into the regulatory mechanism of somatic cell reprogramming. However, the interpretation of the authors' results, especially the genome-wide analyses such as ATAC-seq and RNA-seq, is insufficient to support their conclusions. Therefore, this study is too premature to justify its publication in Nature Communications.

Major Points

1. Some mutants of Sall4 completely inhibit the reprogramming with JGES. Contrary, among the NuRD complex, only gatad2b and chd4 KDs affect the reprogramming, and their effects seem to be weak. Therefore, the importance of the NuRD complex in the reprogramming is not clear.
2. The authors suggested that the SALL4-NuRD complex interaction promotes the reprogramming by silencing the somatic genes. However, some genes that should be upregulated during the reprogramming are repressed by KD of NuRD (ex. Fig. 1g). Moreover, some chromatin loci become repressed and closed in the K5A Sall4 mutant cells (ex Figs. 2i (C4,C5), 3b and 3c). Is there any possibility that the chromatin regions opened by SALL4 and NuRD complex are important for the reprogramming?
3. The explanation of Fig. 2g is confusing. The author described, in p. 4, line 146, "K5A fails to close chromatin in C3 and C6 that highly enriched with AP1, ETS motifs (Fig.2g,2h)." However, the signals in K5A seems to be downregulated and upregulated in C3 and C6, respectively.
4. The interpretations of all PCA analyses are insufficient.
 - In Fig. 1b, the PC1 values do not always become greater in the presence of Sall4. Therefore, the biological meaning of PC1 is unclear.
 - Regarding Fig. 1e, the rationales for the following descriptions are unclear.
Line 102, "delay of reprogramming process with Gatad2b or Chd4 knockdown compared to control"
Line 104, "Interestingly, Gatad2b appears to be a more critical one as evidenced by the slower kinetics of day7 cells in Gatad2b samples compared to day5 cells in control sample."
-In Fig. 4d, the JGE samples seem to get closer to ES cells without Sall4. What is the biological meaning for PC1?

Minor Points

1. In lines 62, 65, and 70, as for "iCD1", "7F", and "iCD3", the authors should clearly state the meanings in the main text.
2. In Fig. 1f, what are the definitions of CO1, CO2, CO3 etc?
3. In Fig. S1c, the definition of FC should be clear.
4. In Fig. S2d, the meaning of "peak number" is unclear.
5. In line 126, the definition of the gradually close region (GCR) is unclear. Therefore, it is hard to understand Figs. S2e-S2g (Extended Data).
6. In Fig2g, the clustering method is unclear. What peaks are used? How do the authors compare the peaks in K5A and WT?
7. In Fig2i, how does this RNAseq data relate to ATACseq?
8. In Fig. S3d, what are "Up with K5A sall4" and "Down with K5A sall4"?
9. Lines 185 and 186, "Fig.3d, lower" and "Fig.3d, upper" are correct?
10. Line 186, the description, "In those two classes of loci, the peak densities are significantly reduced", is unclear. What peaks? How significantly?
11. Line 192, "Among the 9828 loci with elevated level of H3K27ac (Up), 7199 loci lost Gatad2b and 193 loci lost both are selected for further analysis." Is this sentence correct?
12. Line 235, "The WT specific 41264 loci undergo closing in J(N12), enabling the rescue of reprogramming (Fig.4g, 4h)." Are the loci really related to the reprogramming?

Reviewer #3 (Remarks to the Author):

In this manuscript, Wang et al. investigate the mechanisms underlying somatic cell reprogramming induced by the transcription factors Jdp2, Glis1, Esrrb, and Sall4. They initially optimize the culture conditions to render reprogramming highly efficient. Then, they found that the transcription factor Sall4 is essential to induce a pluripotent cell state from somatic fibroblasts. Intriguingly, Sall4 facilitates reprogramming to pluripotency by interacting with components of the NuRD complex. The NuRD complex has been investigated extensively in reprogramming. Yet, it remains unclear how the cooperative interaction between NuRD and different transcription factors remodels the chromatin during the early phases of cell fate reprogramming. The authors provide compelling data showing that Sall4 and NuRD components synergistically bind to chromatin regions and that this interaction is essential for the silencing of the somatic program during reprogramming. Overall, the findings are exciting and of broad interest; however, the manuscript would benefit from additional experiments and some clarifications. I have some major and minor points as indicated below:

1. In figure 1G, depletion of Gatad2b and Chd4 does not affect the silencing of highly expressed MEF genes (cluster 5). However, it significantly alters genes associated with muscle differentiation (cluster 3). This might indicate that MEFs depleted for Gatad2b and Chd4 lose their identity to acquire a muscle fate, thus diverging from the reprogramming trajectory and failing to acquire a pluripotent fate. The authors should assess whether MEFs depleted for Gatad2b and Chd4 are activating canonical muscle proteins and if depletion of NuRD components facilitates the reprogramming of MEFs into myotubes induced by the transcription factor MYOD. This would indicate that NuRD is essential to maintain lineage fidelity, and its disruption leads to the spontaneous acquisition of new fates.

2. The authors should corroborate some of the results shown in Figure 1 by performing flow cytometric analysis of wt and shGatad2b reprogramming intermediates (stain for Oct4GFP, THY-1, SSEA-1). These experiments might shed light on how depletion of NuRD affects the reprogramming trajectory at single-cell resolution.

3. In Figure 2C, the authors should show the overlap as a Venn diagram of the enriched proteins in the IP-mass spec for Sall4 wt and K5A mutant. This analysis will assess whether the K5A mutation is causing a loss of interaction with essential proteins besides the NuRD complex.

4. In the text, the authors mention: "Consistent with the knockdown experiments, K5A fails to close chromatin in C3 and C6 that highly enriched with AP1, ETS motifs (Fig.2g, two h)", yet the cluster 3 shows increased loss of accessibility in the mutant respect to the wt Sall4. The authors should clarify this discrepancy observed across panels 2G, H.

5. In Figure 3, the authors should show gene tracks of genes identified in Figure 3B (differentially acetylated regions).

6. To support the conclusions in Figure 4, the authors should include IP-mass spec data for JDP2, JDP2(N12), ESRRB, and ESRRB (N12). The expectation is that only the JDP2(N12) protein gains interaction with NuRD components. Moreover, the authors should compare the JDP2 and JDP2(N12) ChIP-seq with GATAd2B ChIP-seq data.

Minor points:

- Figures 1F and 1H appear in the text in the wrong order.
- Figure 1F lacks the ESCs data.
- Figure S1F and S1I show the same image for iCD3 and 7F conditions.
- The authors should include viability data for MEFs depleted for Gatad2b/2a and CHD4.
- Figure 4C is missing the reprogramming data with the wt factors.
- The description of the karyotype analysis in the methods section is unclear.
- The legend of FigS5 is unclear.
- Different fonts are used across the text and Figures (Arial and Times). The authors should try to be consistent.

Dear editor and reviewers,

We wish to thank you for your time and effort in reviewing our manuscript and for the constructive comments, which have been very helpful in guiding our revision. We have performed all experiments suggested and revised the manuscript accordingly. We provide this point-by-point rebuttal and hope that you can evaluate this revised manuscript with ease. Please note that the revised texts in manuscript are marked with **red color** and the responses presented in **blue text**.

Main changes and additions made to the Figures.

1. The original fig 1d and fig S2a were combined to revised fig 1d.
2. The original fig S2b and S2c were combined to fig S2a in revised Figure S2.
3. Figure 1f and 1g were exchanged in revised Figure 1
4. The old fig 1h was moved to fig S2b in revised Figure S2
5. We added fig 2e, 2g in the new Figure. The old fig 2h was replaced by a new one.
6. The old fig 2e, 2f were moved to Fig S3a and S3b in new Figure S3.
7. We added fig S3c,3d, 3e,3i, 3j, and 3k to new Figure S3.
8. We added fig 3d, 3e, 3f, 3g, 3h, 3i, 3j, and 3n to new Figure 3.
9. The original fig 3c,3f, 3g, and 3i were moved to Fig S4.
10. We added fig S4a, S4b, S4c, S4e, S4f, S4i, S4j, S4k, and S4m to new Figure S4.
11. We added fig 4c, 4h, 4i, 4j and 4k to the new Figure 4.
12. The original fig 4a and 4d were moved to fig S5a and S5d.
13. We added fig S5b, S5e, S5f to the new Figure S5.

We added an additional Extended Figures to answer critiques from reviewers

More appropriately place current work in context of the literature (i.e. include appropriate citations) (Reviewer #1)

We appreciate this comment and have now carefully revised examined the manuscripts and figures, as well as the cited literatures. All the figures have been arranged in the correct order.

The primary concern of reviewer1 appears to be the concept of somatic chromatin closing. We have now cited the relevant literature and tuned down our claims as suggested by the reviewer. We wish to stress the fact that our initial aim is to show that the well known Sall4-NuRD complex plays a critical role in iPSC generation with the JGES system. We hope that this may further encourage other investigators to reexamine NuRD in reprogramming and other systems of cell fate transitions. It remains to see if this can be extended into other systems. Nevertheless, we feel confident that the work will stand the test of time and can serve to help us understand cell fate control better in the near future.

-Accessibility to all data, including raw data, is mandatory (Reviewer #1, journal policy)

We confirmed that both raw and final processed data have been deposited in a public database.

The data supporting the conclusions of this Article, including CUT&Tag for H3K27ac, Sall4, Jdp2, Gatad2b, Esrrb and Glis1 are available at GEO under accession GSE199612.

The ATAC-seq and RNA-seq data were from GSE199609 and GSE199613. The RNA-seq data of MEF and ES cells was obtained from GSE127927.

Source Data for Figs 1, 2, 4 and Supplementary Fig 1-5 are provided with the manuscript.

Reviewers may use accession numbers GSE199614 and reviewer tokens "oxmrsaeqfluptav" to access the deposited sequencing data. Please visit the datasets <https://www.ncbi.nlm.nih.gov/geo/query/acc.cgi?acc=GSE199614>

-Further explore NuRD independent role of Sall4 (all Reviewers)

We would like to thank the editor and reviewers for their thoughtful comments. To address that, we re-examined the SALL4 and JDP2 IP-MS data and found many proteins (Hmbox1, Tfam, Kpna4, Parp1, and Lig3) are engaged by Sall4, irrespective of K5A mutation (Extended Fig 5a, b). Also, above mentioned proteins are significantly enriched only with the synthetic JDP2^{N12} indicated that the N-terminal region of SALL4 might perform different gene regulatory activities by interacting with alternative proteins (Extended Fig 5c, d). To validate that, knocking down Hmbox1, Tfam, Kpna4, Parp1, and Lig3 by shRNA leads to decreased reprogramming efficiency (Extended Fig 5e, f).

-Clarify experimental procedures/presentation/replicates/etc in detail (all Reviewers)

Thank editor and reviewers for their sincere suggestions.

In the revised manuscript and figures, we reorganized the description of experimental procedures, the methods for the knocking down experiment by shRNA, karyotype, and clustering method for ATAC-seq analysis. All above were sufficiently described.

For presentation, we checked integrity and correctness of all the references, sentences, fonts, figure legends, and images. We provided the word abbreviation as well as the requested list of abbreviations and have labeled the figures in right order. We added definition to CO and OC and detail information for PCA.

We have added required controls, materials and data in the revised manuscript.

-Include appropriate controls/basic characterization (e.g. cell death, cell cycle, etc) (all Reviewers)

We have added required controls, materials, and data, performed a basic characterization (e.g. cell death, cell cycle, etc), and do not see any correlations with the molecular mechanisms.

In cell cycle and cell death analysis, MEFs were infected with retrovirus expressing either luciferase-shRNA or NuRD-shRNA. Then, cell viability was measured by CCK8 assay and the cell cycle was examined by PI staining. Results showed that cells exhibited good cycle stability, and cell viability was not affected after knocking down NuRD or luciferase (Extended Fig 6a-d).

-Provide further, unbiased analysis/presentation of data – e.g. regions with histone acetylation decreases, chromatin regions opened by Sall4; NuRD, etc) (Reviewers #1, #2)

We thank the reviewer for insightful comments.

First, we did not mean to ignore the loci with decreased H3K27ac modification. Since the NuRD complex possesses histone deacetylase activities and is involved in various biological processes including repression of transcription. So, we focused on the increased H3K27ac loci which may lead to the failure of gene inactivation in the K5A cells. Then, transcription factor motif discovery for the gene loci with increased H3K27ac shows that AP-1 family motifs which should be closed during the reprogramming process but remain open. Besides, we analyzed the genes located in H3K27ac increased regions, and show that those genes are involved in angiogenesis and Erk signaling pathways which shows the

inhibitory effect of reprogramming (Extended Fig 7a). Consequently, those results fit our hypothesis that Sall4 recruits NuRD complex to specific loci to regulate transcriptional events during reprogramming.

Meanwhile, we think that the question about gene loci with decreased H3K27ac raised by reviewers is meaningful. To provide unbiased analysis of our sequencing data, we performed comprehensive analysis integrating CUT&Tag, RNA-seq, and ATAC-seq.

First, we examined the biological process of genes isolated from those loci and the chromatin accessibility of those genomic regions. Gene Ontology (GO) analysis shows that genes located in decrease regions are involved in signaling pathways regulating the pluripotency of stem cells (Extended Fig 7b). Second, we assessed the chromatin accessibility dynamics of the decreased regions across the reprogramming process. ATAC-seq analysis revealed that large degree of changes in chromatin accessibility occur from MEF to day1 after JGES induction, then accessibility decrease gradually, and remain higher level in Sall4^{WT} than in Sall4^{K5A}. Besides, increased level of H3K27ac were found in Sall4^{WT} at many pluripotent gene loci (Sox2, Tead4, and Wnt6) (Supplementary Fig.4a, b, c).

These data suggest that stable interactions between Sall4 and NuRD complex are required for silence of somatic program and subsequent activation of pluripotent program.

-Further analyze acetylation changes related to ATACseq data (Reviewer #1)

All requested analyze were performed and results were added in revised figures (Fig 3d, e, f, n, Supplementary Fig S4a, b, c).

-Clarify analysis of RNAseq/PCA (Reviewer #2)

We added clearly interpretation about the RNAseq/PCA in revised manuscript.

-Confirm whether NuRD stabilizes terminal fate (Reviewer #3)

To address whether NuRD complex affect the stability of ESCs and MEFs, we make lentivirus plasmids pLKO.1-shRNA for knocking down NuRD subunits in ESCs (Extended. Fig 2a, b). After infected by shRNA for 2 days and puromycin selection for another 2 days, ESCs showed negligible morphological changes between shScramble and shNuRD, and comparable level of pluripotent gene such as Oct4, Sox2, and Nanog. To exam whether Gatad2a/2b and Chd4 depletion compromised MEFs viability, equal number of MEFs were seeded and infected by retrovirus shRNA for 2 days and another 2 days for puromycin selection, MEFs showed similar morphology and comparable cell number at day4 (Extended. Fig 3a, b). Together, these results suggesting that depletion of NuRD subunits have imperceptible effect for pluripotency of ESCs and negligible for the viability for MEFs.

-Provide additional support for NuRD specific rescue with N12-Jdp2 (Reviewer #3)

We thank the reviewers for their insightful suggestions and agree that IP-MS was helpful in demonstrating that Jdp2^{N12} rescues the Sall4^{K5A} defect during reprogramming. Therefore, we performed Jdp2^{N12} and Jdp2^{WT} specific IP-MS during JGES reprogramming. Proteomic data showed that most NuRD subunits were significantly enriched only by Jdp2^{N12} but not by Jdp2^{WT}, suggesting that the NuRD complex act as a partner of Jdp2^{N12} that exerts gene regulatory activities (Fig. 5c).

However, we did not perform Esrrb IP-MS because Esrrb^{N12} could not rescue Sall4^{K5A} defect during reprogramming. According to motif enrichment by CUT&Tag analysis, ESRRB shows its genomic occupancy to pluripotent loci (Fig S5a). In contrast, CUT& Tag analysis shows that Jdp2 targets genomic loci enriched for somatic TFs such as Fra1, Fos, and Jun-AP1, indicating that Jdp2 acts as a repressor to close chromatin (Fig S5a). So Jdp2 may benefit from N12 grafting, but Esrrb does not. Therefore, we decided not to perform Esrrb-IP-MS experiments.

To validate Jdp2N12 may recruit NuRD complex to orchestrate chromatin remodeling and trigger somatic program inactivation. we performed GATAD2B CUT&Tag experiment during Jdp2N12 and Jdp2WT reprogramming in combination with Esrrb, Glis1, and Sall4^{K5A} on day1. First, we categorized the JDP2N12 and JDP2WT CUT&Tag peaks into the simplest tier of JDP2-Common, JDP2N12 specific, and JDP2WT specific (Fig 4d), and then we analysis of GATAD2B binding density for above three regions during Jdp2WT and Jdp2N12 reprogramming (Fig.4j). For JDP2N12-specific regions, where we observed higher GATAD2B binding density in Jdp2N12 reprogramming. Conversely, the JDP2WT-specific regions exhibit increase GATAD2B occupancy in Jdp2WT reprogramming.

REVIEWER

COMMENTS

Reviewer #1 (Remarks to the Author):

C1: The authors identify Sall4 as being a particularly important component in their alternative remodelling set of transcription factors so they ask what its molecular interactors are by performing an IP-Mass Spec experiment. They identify a well-known Sall4 interactor, NuRD, and then characterise what that might be doing. They perform extensive RNAseq, ChIP-seq and ATAC-seq on reprogramming cells, but rather than use this to determine an order of events for the reprogramming process and the role of Sall4 in it, the data are mined to find evidence to support their model, which is that Sall4 recruits NuRD to silence things. In the end they identify 16 genes at which all of the data indicate this is exactly what is happening. So, while what they say appears to be true for these loci, there are hundreds to thousands of other loci for which their evidence does not support the model. Is everything else that Sall4 does really irrelevant?

R1: We appreciate this comment very much. Indeed, it has been a “biased” approach we took to singly focus on Sall4, and in the process, apparently ignoring other perhaps more important events such as the one mentioned by the reviewer, i.e., order of events during reprogramming. We wished to justify this bias based on the following considerations: 1) in our previous work published in CELL Reports, we documented quite carefully the global changes during 7F reprogramming. The 4 F system more or less follows a similar order of events, so we decided not to pursue that route in presenting this paper; 2) the indispensable role of Sall4 sparked our interest early and encouraged us to pursue its role and mechanistic contribution further; 3) the Sall4-NuRD axis, although previously characterized elsewhere, has not been explored in the reprogramming/ cell fate context. Given the conflicting role of NuRD in the literature, we decided to focus on this story. We realized that we could have been more comprehensive in presenting our vast datasets, but nevertheless decided to focus on this relatively simple story. So, hopefully, our “biased” decision would not distract the reviewer from this interesting axis. We are continuing to explore additional events and functions associated with JGES and will write them up when we believe we have a plausible story in the near future. It is also our hope that this work may inspire other investigators to use this system in their systems of inquiry, especially reprogramming.

We also agree with the reviewer that the 16 genes identified from our sequencing data may not be the entire story as the reviewer pointed out that we did not provide a clear explanation of the correlation between Sall4 and the state of hundreds to thousands of other loci. We have addressed this issue in three separate ways:

First, the 16 genes are important but do not represent all the transcriptional outcomes of Sall4 mutation. We identified the 16 candidates with the following three standards: (1) the genes loci without GATAD2B binding in Sall4^{K5A}, (2) the genes upregulated in Sall4^{K5A} (fig 2i Cluster3), (3) the genes activated in shGatad2b (fig 1f, Group 1) at reprogramming day1. The candidates meet above criterions are regulated by Gatad2b, but not the NuRD complex. Most importantly, when we overexpressed 15 of the candidates during JGES reprogramming, 11 genes especially BMP4 leads to significantly decreased reprogramming efficiency (Supplementary Fig.4m). Additional candidates may be identified by varying the selection criteria and may be tested experimentally as well.

Second, though we have a high knockdown efficiency, residual level of 5%-15% Gatad2b remains during reprogramming. The incomplete deletion of Gatad2b is a significant issue that may fail to ablate the interaction between NuRD and Sall4. Consequently, when we analyzed the commonly upregulated genes

from Sall4^{K5A} and shGatad2b during reprogramming, the list was narrowed down by the insufficient knockdown effect.

Furthermore, genome-wide CUT&Tag analysis of SALL4 binding in both WT and K5A reprogramming identified a total of 19000 regions with increased or decreased H3K27ac modification in K5A. About 2229 genes and 2601 genes are found from above increased and decreased regions, respectively. There are 381 genes from decreased region and 361 genes from increased regions also be found from different expressed genes of Sall4 K5A RNA-seq data. Gene ontology analysis revealed that overlap genes from cluster 3 are associated with extracellular matrix organization, cell adhesion, and wound healing. By analysis the ATAC-seq data, there are 9344 sites show distinct chromatin accessibility dynamics between wild type and K5A mutant Sall4 (fig 2g). Motif analysis show that DNA motifs of the AP-1 and ETS families of TFs were enriched from K5A mutant Sall4 reprogramming. These TFs were detrimental for reprogramming. These results suggest that Sall4 acts in a collaborative fashion with the NuRD complex to reconfigure nuclear architecture in order to trigger cellular reprogramming.

Lastly, there are additional roles for Sall4 besides interacting with NuRD. Minimally, we have shown in the revised figures that DNA repair components are also important for JGES reprogramming. But we did not pursue their mechanism in the manuscript further. We revised part of the text and quoted here as “Unlike NuRD, we also show, through unbiased analysis of the proteomics data, that Sall4^{K5A} and Sall4^{WT} also share many protein partners (Fig.2e). Gene ontology analysis show that the common proteins are associated with DNA repair (Supplementary Fig.3c). To validate the function of those proteins, a decrease in reprogramming efficiency was observed when we knocked down Hmbox1, Tfam, Parp1, Lig3, and Kpna4 by shRNA (Supplementary Fig.3d, e), suggesting that Sall4 engages many partners besides NuRD to facilitate reprogramming. Yet since Sall4^{K5A} is almost totally ineffective, we conclude that the NuRD-Sall4 axis plays a dominant role in JGES reprogramming.” We would not be surprised if Sall4 plays additional role than even DNA repair as it is such a pleiotropic factor in early development and also possess many interesting domains and modifications. Indeed, we are pursuing some of them, especially the role of phosphorylation on Sall4 impacting its function.

C2: An alternate approach would be to identify any previously unknown interactors and ask whether they contribute to Sall4 function in reprogramming. The authors' data show that Sall4 likely has further functions outside of the well-trodden path of NuRD interaction: there are considerably more iPS colonies formed when knocking down Gatad2b or Chd4 than if Sall4 is left out of the reprogramming mix. Even the K5A mutation produces more colonies than if Sall4 is left out. Does this mean that Sall4 also has a NuRD-independent function? If so, what is it? Perhaps the proteomics data can provide some clue?

R2: we again appreciate this critically insightful suggestion. As mentioned above, we have indeed performed such investigation. We also agree with the reviewer that Sall4 also has a NuRD-independent function. It has been reported that SALL4 can interact with many proteins such as NuRD complex, MLL complex, DNMTs and TETs, which define its molecular function (PMID: 26892498, 24051379, 22128185, 34732693). Beside interacting with various epigenetic modulators, the multiple zinc finger transcription factor Sall4 has a unique role in regulating downstream target gene expression by its zinc finger clusters (ZFC). The C-terminal ZFC4 of Sall4 is both necessary and sufficient for its binding to AT-rich motifs located in heterochromatin (PMID: 33406418, 33406384). Deletion of ZFC2/3/4 will significantly compromise Oct4-GFP positive iPS generation (fig S3d). The cysteine residue 420 is essential for Sall4 binding to 5hmC-containing DNA and then cooperate with Tet2 to facilitates further oxidation (PMID: 27840027). The alteration of cysteine to alanine in 420 of Sall4 results in remarkably decline of iPS formation (fig S3b). Those results suggests that in addition to NuRD-independent function, the DNA ability of Sall4 is pivotal to promote somatic cell reprogramming.

As discussed in the earlier section, we have explored further the SALL4 IP-MS data to identify NuRD-independent function. According to unbiased analysis of the proteomics data, Sall4^{K5A} and Sall4^{WT} were highly overlapping among the IP-MS enriched proteins (Fig.2e). Gene ontology analysis showed that the common proteins were associated with DNA repair (Supplementary Fig.3c). To validate the function of those proteins, a decrease in reprogramming efficiency was observed when we knocked down Hmbox1, Tfam, Parp1, Lig3, and Kpna4 by shRNA (Supplementary Fig.3d, e). These have all been revised into the new version.

Specific

comments:

C3: The author suggest in the Abstract that “Selective closing of open chromatin may be an early step in cell fate decision.” This does not seem like a new idea to me. The first time I can recall this being highlighted for reprogramming was in the following publication, but I suspect there have been many other papers as well: Chronis C, Fiziev P, Papp B, et al. Cooperative Binding of Transcription Factors Orchestrates Reprogramming. *Cell*. 2017 Jan;168(3):442-459.e20. DOI: 10.1016/j.cell.2016.12.016. PMID: 28111071.

R3: In Chronis paper, they reported that three reprogramming transcription factors (TFs) OCT4, SOX2, and KLF4 (OSK) first engage active somatic enhancers to initiate their silencing by redistributing somatic TFs and decommissioning enhancers. However, they did not investigate how they close and silence somatic enhancers directly. In our own publication, Li et al *Cell Stem Cell* 2017, we demonstrated the concept of chromatin closing and show that Sap30 is activated by OSK and then to recruit Sin3A to close open chromatin. So, in this sense, this concept is no longer new anymore. We have revised this statement as the following “These results identify a previously unrecognized role of NuRD in reprogramming, and may further illuminate chromatin closing as a critical step in cell fate control”. We can also delete this statement if the reviewer sees as more appropriate. Nevertheless, with JGES, we provided here a mechanism on how somatic open chromatin loci are closed with the Sall4-NuRD complex, demonstrating that diverse mechanisms may be utilized by cells to close initial open chromatin during cell fate transitions. We are not aware of other papers at this point, but would be very happy to include those once we found them.

C4: Figure 1c shows that they have identified a large number of interactors. Prominent amongst them are NuRD components. Sall4 and NuRD are well known to co-purify in a range of cell types, so this result is expected, but it is reported in the text as if this were a novel finding.

R4: We appreciate this comment. When we started the experiments, we did not expect to see NuRD, perhaps due to very very limited knowledge in this field. Then, we realized the wealth of information on NuRD-Sall4 interaction and have cited some of those in the text and also discussion. We have revised the early part as well to reflect the earlier discoveries.

C5: Figure 1c shows a large number of interactors, but only two which are not NuRD components (Parp, Lig3) are labelled. There is no indication of why these two proteins are also labelled, unless they have been mistaken for NuRD components? What are all of these other interactors? How many also purify with the K5A mutant? These proteomics data are not made available, nor have the authors indicated that they have deposited these data in any repositories. There is no mention in the Methods of how these experiments were performed.

R5: We apologise these confusing points. In detail, among the Sall4 and IgG enriched proteins, we selected the top 8 interactors and labeled them in figure 1C. Parp1 and Lig3 were among the top 8 enriched proteins.

To avoid confusion, in revised Figure 1c, only the NuRD subunits and Sall4 itself were labeled among Sall4 enriched proteins. Gene ontology analysis about Sall4 and IgG enriched proteins were listed below.

We performed further analysis about the proteins that purified by Sall4^{WT}, Sall4^{K5A}, and both. Among all Sall4^{WT} enriched targets, there are 44 proteins also purified by Sall4^{K5A} (Fig 2e). Gene ontology analysis about genes from above three groups were listed in revised fig S2c.

We added an Excel termed Supplementary Data for the requested proteomic information. All detail about the SALL4 IP-MS, JDP2 IP-MS, and overlap analysis of Sall4^{N12} and Sall4^{K5A} could be found in those three sheets.

C6: There is no description of how the knockdowns were done or what the timelines are. Knocking down/out Chd4 and Gatad2a/b has been reported to result in cells undergoing cell cycle arrest: do the authors know that failure to reprogramme is not simply caused by the cells exiting the cell cycle?

R6: We appreciate these comments and offer the following description. Knocking down genes are routinely performed in the lab. In this case, we made pSuper-Puro retrovirus vectors (addgene) containing shRNA sequences for the gene of interest in our lab. The target sequence for each shRNA was designed online (Sigma). pSuper-shRNA was constructed using the enzyme-digested technique.

The knockdown efficiency of each shRNA was measured during JGES reprogramming. To produce infectious retroviral particles, PlatE cells cultured on 6 cm dishes were transfected with target shRNA. Viral supernatants were harvested after a 2-day transfection. The supernatant was administered in MEF culture together with JGES retrovirus. The volume of the shRNA virus is the same as that of one factor in JGES. After two infections and culture for another 48 hours, cells were selected in puromycin for 3 days. After all the negative control cells infected by JGES died, another shRNA sample was harvested for RNA isolation. The knockdown efficiency was measured in real-time PCR.

Then the selected pSuper-shRNA and reprogramming factor pMX-Jdp2/Glis1/Esrrb/Sall4 were transfected into PlatE cells for retrovirus preparation separately. MEFs were infected by shRNA and JGES retrovirus simultaneously twice. Puromycin selection was performed between day2 and day5. Oct4-GFP positive colonies were counted on day7.

We chose Cell Counting Kit-8 (CCK-8) to measure the cell viability and proliferation when knockdown Chd4 and Gatad2a/b (Extended. Fig 6a). The results show that knockdown neither Chd4 nor Gatad2a/b results in cell cycle arrest. Cell cycle measurement by propidium iodide (PI) staining was performed by flow cytometry and results showed that cell cycle was not affected by depletion of NuRD complex (Extended. Fig 6b-d). Based on that result, we conclude that the decrease in reprogramming efficiency with Gatad2b/2a/Chd4 knockdown was not related to higher proliferation or cell cycle arrest.

C7: Figure 1d, extended fig 2: The authors are stating the knocking down Chd4 and Gatad2 proteins has the most pronounced effect on reprogramming. The point is to be able to compare the effects for different components. so the data in these two figures should be combined and shown in one main fig: (Extended fig 2a has a typo in the figure: “Kncok NuRD Subunits”)

R7: Thanks for pointing out this. Now, we have combined the original fig 1d and fig S2a as revised fig 1d. The original fig S2b and S2c were combined as fig S2a in revised Figure S2. We have corrected the typo.

C8: Figure 1e: very difficult to see the data. Appears to be two replicates, but I can't see the final Chd4 KD datapoint. It looks like WT and Chd4 kd are about the same at D7...what do we conclude

from this? How were the arrows drawn in?

R8: we again wish to thank the reviewer for the comments. PCA tools can reduce the dimensionality of data through principal component analysis, and view principal component related features at a two-dimensional level. We performed RNA-seq by collecting two replicates of each sample of MEFs, reprogramming Day0/1/3/5/7, and five replicates of ESCs. To minimize confusion, we have modified the figure and hope that it is clear to check the replicates and data points. A zoomed-in snapshot of all day5 and day7 samples were placed at the bottom of the PCA map. There is an almost complete overlap between one of the Chd4 and Gatad2b KD datapoints at day7 as shown in the middle of the snapshot.

PCA shows that the gene transcription landscape as assessed by RNA-seq undergoes gradual transitions that bridge between those of MEFs and ESCs. We further compare the transcriptomes among shLuciferase, shChd4, and shGatad2b and show that there is a delay of transition with Gatad2b (Yellow) or Chd4 (Blue) knockdown compared to control (Green) in their transcription programs to arrive at pluripotent state. Though one of the shChd4 sample at day7 looks like shLuciferase, another shChd4 sample and two shGatad2b samples on day7 show further distance to ESCs than shLuciferase on day5. Those analyses suggest that both Chd4 and Gatad2b are required for JGES reprogramming. The curve with arrows was drawn to show the roadmap from MEF, day0/1/3/5/7, to ESC.

C9: Line 110: the authors conclude that their data “suggested that Sall4 recruit NuRD complex to silence the somatic gene.” So far they show two perturbations which inhibit reprogramming: leaving Sall4 out of the mix, and knocking down NuRD components. In both cases there will be a failure to progress down the reprogramming trajectory in terms of gene expression, chromatin changes, etc. This does not mean that the two things are necessarily working together, just that they both fail to reprogramme.

R9: Our IP-MS experiment shows high-affinity interaction between Sall4 and almost all NuRD components. When we analyzed the overlapping genes perturbed by Sall4 dropout and NuRD components knockdown and showed that a cooperative role between them to repress somatic gene expression. However, we agree that we should not make such a strong statement at this point. We revised it as “These results suggest that Sall4 may mediate reprogramming through components of the NuRD complex”.

”

C10: The data with the K5A mutation argues very strongly that the majority of the effect they are showing relies on a Sall4-NuRD interaction. They show through mutational analysis that most of the zinc fingers also play a contributing role to Sall4 function in reprogramming (assuming none of these impact protein stability), but this is not explored further.

R10: This study, as pointed out by the reviewer correctly, is about the Sall4-NuRD axis. So, we did not spend too much time to explain additional findings we made. Nevertheless, we have performed additional experiments and obtained interesting results regarding other parts of Sall4 such as the zinc fingers which have been reported in other context such as PMID: 17295837, 16443351, 33406384, 16790473. We thought that those Investigations are beyond the scope of our paper, which aims to show that the

interaction between Sall4 and NuRD is key for somatic gene silencing. However, we are pursuing them and would like to report them if any new insights can be obtained mechanistically. We apologise not being comprehensive on Sall4, but just narrowly focusing on the Sall4-NuRD interactions at this point.

C11: Lines 151-156: they specify that K5A cells resemble shGatad2b cells. Does this imply that they don't resemble shChd4 cells? Any idea why not?

R11: We performed Gene Set Enrichment Analysis (GSEA) to determine whether statistically significant difference between Sall4^{K5A} mutation and NuRD subunits knockdown. First, a list of ranked gene set was made based on differential expression between Sall4^{WT} and Sall4^{K5A}. Then an enrichment analysis was performed to determine the statistically difference between genes from each group (fig 1f) and the ranked gene set. Since each group was identified by different express genes with similar pattern either from shGatad2b or shChd4, GSEA results should represent the association between Sall4^{K5A} mutation and both shGatad2b and shChd4. However, shChd4 was unintentionally left out from our conclusion, so we fixed this and the original sentence was replaced by the following sentence: "Coincidentally, those GO terms observed in Sall4^{K5A} groups can also be found after depletion of Gatad2b and Chd4 during JGES reprogramming (Fig.1f). Consistently, by comparing the RNA-seq data in each group (Fig 1f) with Sall4^{K5A} upregulated or downregulated gene sets, we show statistically significant concordance between shGatad2b/Chd4 and Sall4^{K5A} (Supplementary Fig.3l)."

C12: Line 169: 14.5% of gene loci show an increase in K27Ac in the K5A cells vs WT and they focus on these. Yet there is no justification given for why they ignore the very similar number of loci showing a decrease in K27Ac, other than the former examples fit their model.

R12: We thank the reviewer for insightful comments. First, we did not mean to ignore the loci with decreased H3K27ac modification. Since the NuRD complex possesses histone deacetylase activities and is involved in various biological processes including repression of transcription. So, we focused on the increased H3K27ac loci which may lead to the failure of gene inactivation in the K5A cells. Then, transcription factor motif discovery from the gene loci with increased H3K27ac shows that AP-1 family motifs which should be closed along the reprogramming process but were enriched. Besides, we analyzed the genes located in H3K27ac increased regions, and show that those genes are involved in PI3K-Akt and MAPK signaling pathways which shows the inhibitory effect of reprogramming. Consequently, those results support our hypothesis that Sall4 recruits NuRD complex to specific loci to regulate transcriptional events during reprogramming.

Meanwhile, we think that the question about gene loci with decreased H3K27ac raised by reviewers is meaningful. For decrease regions, decreased level of H3K27ac and chromatin accessibility were found in Sall4^{K5A} and reduced H3K27ac was found at many pluripotent gene loci (Sox2, Tead4, and Wnt6) (Supplementary Fig.4a, b, c). However, we are not sure how mechanistically Sall4-NuRD can impact this category of genes/loci. In the revised text, we added description of this part.

C13: Lines 174-175: the authors conclude that Sall4 normally recruits NuRD to K27Ac-high regions to close them. Yet they only do K27Ac ChIP for one time point: Day1. So we don't know that these sites show any change in K27Ac normally across the time course. How do they behave across the ATAC-seq time course? Do they get closed?

R13: For regions where H3K27ac was increased in Sall4 K5A at day1, we observed a gradual decrease in both H3K27ac and chromatin accessibility across the time course. Of note, compared to Sall4 WT, not only H3K27ac but chromatin accessibility shows elevated level in most Sall4 K5A samples. This strengthens the model that Sall4 recruit NuRD to reduce H3K27ac and to close chromatin.

C14: Lines 176-177: “NuRD contains histone deacetylases capable of removing H3K27ac, resulting in nucleosome compaction and gene silencing” NuRD-mediated nucleosome compaction and gene silencing has been shown to precede H3K27 deacetylation in two different studies. I am not aware of any studies showing that H3K27Ac results in compaction and gene silencing.
R14: We thank the reviewer for pointing this out. Indeed, the erasure of H3k27 acetylation is not directly related to chromatin compaction or gene silencing. We modified the sentence as the following: NuRD complex cooperates with transcription factors to regulate gene expression at the level of chromatin by ATP-dependent chromatin remodeling and histone deacetylases activity. The modification of histone tails such as H3K27ac is tightly coupled to chromatin accessibility and gene expression. Thus, dissociation between the NuRD complex and Sall4 K5A can result in the failure of NuRD-mediated deacetylation of histone H3K27 at somatic gene loci.

C15: Lines 198-199: This is quite a small number of genes. When comparing datasets like this there will always be overlaps, so how do we know this really is significant? Perhaps a more relevant question is why don't we see a bigger effect if Sall4 is recruiting NuRD to repress genes? If this is really an important mechanism then why can they only find 16 genes that support their model?
R15: Thanks for reviewer's comments. Please refer to Response 1.

C16: Line 257: They state that the Sall4-NuRD connection has been reported in cancer. While true it's also been reported in many other cell types.
R16: We have made the change. The new sentence reads as follows “Sall4-NuRD interaction has been reported in cancer and in development processes such as hematopoiesis and neurogenesis”

C17: Reference 27 is incomplete.
R17: We thank the reviewer for pointing this out and the requested reference has been added

Reviewer #2 (Remarks to the Author):

The somatic cell reprogramming process is suitable for studying cell fate control. In this study, the authors demonstrated that mouse embryonic fibroblasts (MEFs) could be reprogrammed with high efficiency by introducing Jdp2, Glis1, Esrrb and Sall4 (JGES) and culturing in a specific chemically defined medium (iCD3) and that the Sall4-NuRD interaction is essential for the reprogramming with JGES. Moreover, they want to show that Sall4 recruits the NuRD complex to the open chromatin regions, including many somatic genes, leading to closing the chromatin regions to enhance the initiation event in the reprogramming.

This study is potentially interesting because it might provide us with new insight into the

regulatory mechanism of somatic cell reprogramming. However, the interpretation of the authors' results, especially the genome-wide analyses such as ATAC-seq and RNA-seq, is insufficient to support their conclusions. Therefore, this study is too premature to justify its publication in Nature Communications.

Major

Points

C1. Some mutants of Sall4 completely inhibit the reprogramming with JGES. Contrary, among the NuRD complex, only gatad2b and chd4 KDs affect the reprogramming, and their effects seem to be weak. Therefore, the importance of the NuRD complex in the reprogramming is not clear.

R1: We appreciate this insightful comment. Indeed, the KD results were very troubling for us initially. The weak effects observed for components of NuRD remain unresolved at this point. There appears to be a consistent pattern for chromatin remodellers, namely the BAF, Sin3A in addition to NuRD. In our previous publications, we have dealt with BAF and Sin3A before, and they all show only partial results with KDs. We are continuing to investigate this aspect of these important complexes. We wish to develop PROTAC type of methods to deplete them and see if we can achieve much more robust results.

For precise the same reason, we decided to focus on the interactions between Sall4-NuRD and show much more interpretable results as pointed out by the reviewer. With these mutant data, we would not be able to write such a manuscript. As such, we decided to focus on the Sall4-NuRD axis.

So the KD results serve as a prelude to our mutation experiments and also the partial rescue with N12-Jpd2. We thought that the rescue experiments validate our model nicely.

Nevertheless, we were thinking along the following logic: First, though we have a high knockdown efficiency, about 5%-15% of Gatad2b and Chd4 remain in cells during reprogramming. The incomplete deletion of Gatad2b or Chd4 is a significant issue that may fail to diminish the interaction between NuRD and Sall4. Second, the subunits that built the NuRD complex cooperate to regulate gene expression and a single subunit knockdown may not achieve a strong effect. But, the K5A mutant that separates Sall4 itself from the NuRD complex totally inhibits reprogramming. Mechanism study by integrating analysis of CUT&Tag and ATAC-seq data reveals that Sall4 cooperates with the NuRD complex to inhibit chromatin accessibility at MEF-specific genes. Hence, taking together the evidence we have provided, we believe that the NuRD complex is a critical co-factor for the repression of select Sall4 target genes during JGES reprogramming.

2. The authors suggested that the SALL4-NuRD complex interaction promotes the reprogramming by silencing the somatic genes. However, some genes that should be upregulated during the reprogramming are repressed by KD of NuRD (ex. Fig. 1g). Moreover, some chromatin loci become repressed and closed in the K5A Sall4 mutant cells (ex Figs. 2i (C4, C5), 3b and 3c). Is there any possibility that the chromatin regions opened by SALL4 and NuRD complex are important for the reprogramming?

R2: We appreciate this comment. We can not rule out completely the possibility that Sall4-NuRD opens chromatin important for reprogramming. However, we have not obtained any direct evidence for this.

As Reviewer mentioned that some genes should be upregulated during reprogramming but are repressed by KD of NuRD or Sall4 mutant defective in the interaction with NuRD, we believe that those genes are activated in a later phase during reprogramming after the initial indirectly. Besides, genes from C5 of Fig2i that failed to be activated in Sall4^{K5A} mutant cells are not expressed in ESC, therefore may not be important for reprogramming. In addition, RNA-seq data of fig 2k (C3) show that lots of genes that failed to be repressed are involved in MEF-related terms of extracellular matrix organization, cell adhesion, and fibril organization. Further analysis of Sall4 and H3K27ac CUT&Tag data shows a group of 2229 genes located in regions with increased H3K27ac modification. Combining analysis of RNA-seq with CUT&Tag data shows that the common genes are involved in somatic specific terms of angiogenesis, wound healing, and cell adhesion. Furthermore, analysis of chromatin accessibility of the H3K27ac increased regions showed that those regions are enriched with motifs for AP-1 family TFs. Based on these results, we believe that the primary function of Sall4-NuRD complex is to close somatic chromatin loci which is important for reprogramming at early stage.

For decrease regions, increase level of H3K27ac and chromatin accessibility were found in Sall4^{WT} and elevated H3K27ac was found at many pluripotent gene loci (Sox2, Tead4, and Wnt6) (Supplementary Fig.4a, b, c), suggesting some chromatin regions maybe opened by Sall4 and NuRD complex, perhaps in an indirect way. This complex issue may be resolved in our future studies.

3. The explanation of Fig. 2g is confusing. The author described, in p. 4, line 146, "K5A fails to close chromatin in C3 and C6 that highly enriched with AP1, ETS motifs (Fig.2g,2h)." However, the signals in K5A seems to be downregulated and upregulated in C3 and C6, respectively.

R3: We are sorry that Fig 2g in the original manuscript was somewhat difficult to interpret. To address this, we performed motif enrichment for ATAC seq and the results have been added to the revised Figure

2h. The new sentence was added as follows: A total of 9344 ATAC-seq peaks can be classified into 6 clusters according to chromatin accessibility (Fig.2f). Among the 6 clusters, more than 2/3 of regions are open in Sall4^{K5A} but closed in Sall4^{WT} (C4, n=4573 and C6, n=1904). Besides, there are two interesting clusters that exhibit a loss of accessibility in Sall4^{K5A} but become accessible (C5) and inaccessible (C3) progressively in Sall4^{WT}, respectively.

Motif enrichment for each cluster shows that enriched motifs are quite different between Sall4^{WT} and Sall4^{K5A}. For example, motifs from ETS (ETS1) and HOMEBOX family (Lhx3, Lhx1, Dlx1, Dlx3) members are specifically enriched in C6 and C4, respectively. Motifs for ETS and FOX family (FoxK2, FoxO3) members are both found in C6 and C4. Moreover, motifs for TFs from the AP-1 family such as Fosl2, Fra1/2, c-Jun, and JunB were also found in C6. Previous findings suggest that refractory cells fail to lose chromatin accessibility and TFs from AP-1 and ETS family members are barriers to reprogramming (PMID:28111071, 34181046, 29220666). Furthermore, overexpression of FoxK2 and FoxO3 significantly inhibit iPSC induction (PMID: 34212295). Lhx3 and Dlx1/2 selectively drive fibroblast to distinct subtypes of neurons (PMID:28886366, 27939218, 25374357, 34592167). On the other, motif enrichment for C5 shows that pluripotent TFs such as OCT4/6, KLF4, and SOX17/21 are only found in Sall4^{WT} but not in Sall4^{K5A}. Those results suggest that the interaction between Sall4 and NuRD complex is required to reconfigure the chromatin architecture necessary for reprogramming.

4. The interpretations of all PCA analyses are insufficient.

R4: We thank the reviewer for pointing this out. We have revised those in manuscript.

5 In Fig. 1b, the PC1 values do not always become greater in the presence of Sall4. Therefore, the biological meaning of PC1 is unclear.

R5: The PC1 represents transcriptome features at different stage of reprogramming. The difference of PC1 values become greater without or with Sall4, thus suggests that Sall4 is the key factor differentiating those two samples. Moreover, MEFs infected with Sall4 shows similar PC1 values. Those results suggest that Sall4 is the main force driving MEFs to ES cells.

6 Regarding Fig. 1e, the rationales for the following descriptions are unclear. Line 102, "delay of reprogramming process with Gatad2b or Chd4 knockdown compared to control"

Line 104, "Interestingly, Gatad2b appears to be a more critical one as evidenced by the slower kinetics of day7 cells in Gatad2b samples compared to day5 cells in control sample."

R6: we again wish to thank the reviewer for the comments. PCA tools can reduce the dimensionality of data through principal component analysis, and view principal component related features at a two-dimensional level. We performed RNA-seq by collecting two replicates of each sample of MEFs, reprogramming Day0/1/3/5/7, and five replicates of ESCs. To minimize confusion, we have modified the figure and hope that it is clear to check the replicates and data points. A zoomed-in snapshot of all day5 and day7 samples were placed at the bottom of the PCA map. There is an almost complete overlap between one of the Chd4 and Gatad2b KD datapoints at day7 as shown in the middle of the snapshot.

PCA shows that the gene transcription landscape as assessed by RNA-seq undergoes gradual transitions that bridge between those of MEFs and ESCs. We further compare the transcriptomes among shLuciferase, shChd4, and shGatad2b and show that there is a delay of transition with Gatad2b (Yellow) or Chd4 (Blue) knockdown compared to control (Green) in their transcription programs to arrive at pluripotent state. Though one of the shChd4 sample at day7 looks like shLuciferase, another shChd4 sample and two shGatad2b samples on day7 show further distance to ESCs than shLuciferase on day5. Those analyses suggest that both Chd4 and Gatad2b are required for JGES reprogramming. The curve with arrows was drawn to show the roadmap from MEF, day0/1/3/5/7, to ESC.

7 In Fig. 4d, the JGE samples seem to get closer to ES cells without Sall4. What is the biological meaning for PC1? R7: we respectfully disagree with the reviewer's interpretation of Fig 4d. From PC1 view, JGE shows the furthest distance to ES cells, even JGES^{K5A} become closer to ES cells than JGE. The PC1 represent transcriptome features at different stage of reprogramming towards pluripotency. As shown above, the ESCs are around ~150 in PC1 compared to MEF around ~-40 on average.

Minor Points

1. In lines 62, 65, and 70, as for "iCD1", "7F", and "iCD3", the authors should clearly state the meanings in the main text.

R1: The requested annotation has been added to the main text. 7F is indicated as Jdp2, Esrrb, Sall4, Nanog, Glis1, Kdm2b, Mkk6. iCD1 and iCD3 refers as iPS Chemically Defined medium1 and medium3, respectively.

2. In Fig. 1f, what are the definitions of CO1, CO2, CO3 etc?

R2: The CO and OC peaks are defined based on the day of opening and closing, covering the changes in chromatin between MEFs and iPSCs/ESCs. For example, CO1 peaks are the chromatin loci that remain close in MEFs but open up during day0. OC1 means regions that are accessible in MEFs but lose accessibility during day0. We followed the same labelling as we reported in 2017 Cell Stem Cell paper.

3. In Fig. S1c, the definition of FC should be clear.

R3: We have made that change. FC is short for Fold Change, which is calculated with the average of Oct4-GFP positive colonies numbers on day7 by dividing the treatment with chemical by control (treat with DMSO). All data has three repeats. FC=1.5, p=0.05 are used as threshold.

4. In Fig. S2d, the meaning of "peak number" is unclear.

R4: The requested annotation has been added. Fig. S2c shows a number of different regions which changes in a different mode (begin open/close at a different time). The data is shown as a heatmap in Fig. 1g. As all regions were identified by Genrich, we use "peak" here. Maybe "region" is a better description.

5. In line 126, the definition of the gradually close region (GCR) is unclear. Therefore, it is hard to understand Figs. S2e-S2g (Extended Data).

R5: To define of the Gradually Close Region for each transition path (MEF to ESC), we identified the gradually closed regions by subtracting the normalized ATAC-seq signal between adjacent stages and less than threshold. The gradually closed regions that crossed three transition paths

was subsequently termed as Gradually Close Region. The threshold is set as 0.05 multiplied by the range of the normalized ATAC-seq signal.

6. In Fig2g, the clustering method is unclear. What peaks are used? How do the authors compare the peaks in K5A and WT?

R6: As huge amounts of data in atac-seq data, the clustering method to ATAC-analysis in the paper is minisom, which is a minimalistic and Numpy based implementation of the Self Organizing Maps (SOM). More information of minisom is in <https://github.com/JustGlowing/minisom>.

The analysis method has been described at the end of the article. First, every sample was called for peaks by macs2 with same parameters individually. Second, all peak files will be merged to a peak set file. (If two or more peak has overlap, they will merge to one). last, the bam was used to count reads or RPKM on the region in peak set file. After that, we used the reads for significance analysis by DEseq2. Refer to "DiffBind" for specific analysis algorithm and ideas in <https://bioconductor.org/packages/release/bioc/html/DiffBind.html>.

7. In Fig2i, how does this RNAseq data relate to ATACseq?

R7: In response to the Reviewer's comment, we analyzed the gene expression for the genes closest to the ATAC-seq peaks within each cluster.

First, we identified promoters from regions clustered in ATAC-seq (fig 2f) within 1kb from the TSS. Then genes that were located downstream of promoters were isolated and their expression values are displayed as z scores of normalized counts. We found that changes in transcript abundance broadly correlate with changes in chromatin accessibility. For example, genes isolated from C4 and C6 show higher expression levels in Sall4^{K5A} than Sall4^{WT} during reprogramming, similar patterns were also found in RNA-seq G1 and G6. Conversely, genes isolated from C5 show higher expression levels in Sall4 WT than in Sall4 K5A, similar patterns were observed in RNA-seq G2, G4, and G5.

Moreover, we observed chromatin accessibility dynamics correlate with changes in gene expression at many chromatin loci. For regions that are more accessible in Sall4^{WT}, we observed a higher level of gene expression in Sall4^{WT} than in Sall4^{K5A} such as Mas1, Peg10, and Pkd1. In contrast, for regions in which accessibility is established in Sall4^{K5A} but remains inaccessible in Sall4^{WT}, there was a more significant extent increase in gene expression in Sall4^{K5A} than in Sall4^{WT} such as Bicc1, Fmo1, and Sox5 (Fig S3i, j).

Taken together, these results led us to conclude that gene expression is closely related to chromatin accessibility.

8. In Fig. S3d, what are "Up with K5A sall4" and "Down with K5A sall4"?

R8: "Up/Down with K5A sall4" means the gene expression change (up/down regulation) by comparing JGES^{K5A} to JGES^{WT} on day 3.

9. Lines 185 and 186, "Fig.3d, lower" and "Fig.3d, upper" are correct?

R9: The reviewer is correct and we have fixed the error.

10. Line 186, the description, "In those two classes of loci, the peak densities are significantly reduced", is unclear. What peaks? How significantly?

R10: Among genomic regions occupied by both Sall4^{WT} and Sall4^{K5A}, large numbers of Gatad2b-bound peaks (7199) were identified mainly in WT conditions. A similar Gatad2b binding pattern was also found among Sall4^{WT}-specific regions (463). The weak signals at sites occupied by Gatad2b in the K5A condition reveal that Gatad2b genomic binding ability is severely compromised when Sall4 dissociates with NuRD complex, but nevertheless showing residual binding as indicated.

11. Line 192, "Among the 9828 loci with elevated level of H3K27ac (Up), 7199 loci lost Gatad2b and 193 loci lost both are selected for further analysis." Is this sentence correct?

R11: The reviewer is correct and we adapted this sentence as follows: Integrated analysis of loci that show an elevated level of H3K27ac (9828) with loci that loss Gatad2b binding (7199+463),

we identified a total of 610 common loci and found a reciprocal relationship between Gatad2b and H3K27ac (Fig.3m, green vs red).

12. Line 235, "The WT specific 41264 loci undergo closing in J(N12), enabling the rescue of reprogramming (Fig.4g, 4h)." Are the loci really related to the reprogramming?
R12: Those loci are related to JGES reprogramming. We provided evidence as follows:

First, motif enrichment analysis of those loci shows that motifs for AP-1, ETS, and ATF family TFs are enriched. Our previous paper showed that those TFs impede reprogramming by disrupting chromatin dynamics (Li et al 2017).

Second, chromatin accessibility dynamics correlates with changes in gene expression. we identified genes that are specific to these loci such as MEF-related genes Tgfb2, Htra1, and Bmp1 highly expressed in MEFs and Jdp2^{WT} but reduced in Jdp2^{N12} and ESCs. This is consistent with chromatin accessibility dynamics that those chromatin loci remain open chromatin in MEFs and Jdp2^{WT}, and conversely acquire close chromatin in Jdp2^{N12} and ESCs (Fig 4i).

More importantly, overexpression of Tgfb2, Htra1, and Bmp1 inhibits JGES reprogramming (Fig S4m).

Taken together, those loci are critical for JGES reprogramming.

Reviewer #3 (Remarks to the Author):

In this manuscript, Wang et al. investigate the mechanisms underlying somatic cell reprogramming induced by the transcription factors Jdp2, Glis1, Esrrb, and Sall4. They initially optimize the culture conditions to render reprogramming highly efficient. Then, they found that the transcription factor Sall4 is essential to induce a pluripotent cell state from somatic fibroblasts. Intriguingly, Sall4 facilitates reprogramming to pluripotency by interacting with components of the NuRD complex. The NuRD complex has been investigated extensively in reprogramming. Yet, it remains unclear how the cooperative interaction between NuRD and different transcription factors remodels the chromatin during the early phases of cell fate reprogramming. The authors provide compelling data showing that Sall4 and NuRD components synergistically bind to chromatin regions and that this interaction is essential for the silencing of the somatic program during reprogramming. Overall, the findings are exciting and of broad interest; however, the manuscript would benefit from additional experiments and some clarifications. I have some major and minor points as indicated below:

1. In figure 1G, depletion of Gatad2b and Chd4 does not affect the silencing of highly expressed MEF genes (cluster 5). However, it significantly alters genes associated with muscle differentiation (cluster 3). This might indicate that MEFs depleted for Gatad2b and Chd4 lose their identity to acquire a muscle fate, thus diverging from the reprogramming trajectory and failing to acquire a pluripotent fate. The authors should assess whether MEFs depleted for Gatad2b and Chd4 are activating canonical muscle proteins and if depletion of NuRD components facilitates the reprogramming of MEFs into myotubes induced by the transcription factor MYOD. This would indicate that NuRD is essential to maintain lineage fidelity, and its disruption leads to the spontaneous acquisition of new fates.

R1: We appreciate this insight noticed by the reviewer and indeed checked the genes expressed in those three GO terms related to muscle contraction (Cluster 3). We found that many voltage-gated channel-related genes were upregulated, but canonical muscle genes such as Acta1, Myh11/9, Myom2/3, Tpm1/2/3, and Myod1 remain stable (Extended. Fig 4a-d). To validate that, MEFs depleted for Gatad2b, Chd4, and luciferase were reprogrammed by Myod1 or DsRed. The day3 reprogramming samples were collected for Q-PCR test and results indicating that Myod1 activate Acta1, Myod1, and Tnnt1, irrespective to depletion of NuRD complex. In contrast, the voltage-gated channel-related genes were upregulated by depletion of NuRD complex, irrespective to overexpression of Myod1. Together, depletion of Gatad2b or Chd4 did not facilitate the reprogramming of MEFs into myotubes by Myod.

2. The authors should corroborate some of the results shown in Figure 1 by performing flow cytometric analysis of wt and shGatad2b reprogramming intermediates (stain for Oct4GFP, THY-1, SSEA-1). These experiments might shed light on how depletion of NuRD affects the reprogramming trajectory at single-cell resolution.

R2: Thank the reviewer for the constructive suggestion. We have performed flow cytometric during reprogramming with depletion of Gatad2b, Chd4 and luciferase (Extended Fig 1). There were gradually decrease and increase of Thy1 positive and SSEA1 positive cells along reprogramming, respectively. The proportion of SSEA1 positive cell is much higher with control (luciferase) than depletion of Gatad2b or Chd4.

3. In Figure 2C, the authors should show the overlap as a Venn diagram of the enriched proteins in the IP-mass spec for Sall4 wt and K5A mutant. This analysis will assess whether the K5A mutation is causing a loss of interaction with essential proteins besides the NuRD complex.

R3: we appreciate this suggestion. We performed analysis of the IP-MS data and all the significant proteins were divided into three groups: Sall4 WT specific, Common proteins, and Sall4 K5A specific. Detail information about all the proteins in each group could be found in Supplementary data. Gene ontology analysis of these proteins showed that Sall4 WT specific proteins associated with regulation of cell fate/stem cell specification. Inspection of genes from these GO terms showed they were all NuRD subunits. We agree that it would be useful to identify NuRD-independent proteins for exploring the role of Sall4 during reprogramming. As suggested, we investigate NuRD-independent proteins by focusing on the common proteins and JDP2^{N12} specific enriched proteins (Extended. Fig 5a-f). When Hmbox1, Tfam, Parp1, Lig3, and Kpna4 were knocked down by shRNA, reprogramming efficiency decreased, suggesting these proteins also play critical roles for JGES reprogramming. This means that Sall4 facilitates reprogramming through NuRD dependent as well as independent mechanisms. Since the independent mechanisms seem quite diverse, we maintain our focus on the Sall4-NuRD axis in JGES reprogramming. The independent mechanisms will be studied in the near future and we wish to report them if we identify concrete mechanism.

4. In the text, the authors mention: "Consistent with the knockdown experiments, K5A fails to close chromatin in C3 and C6 that highly enriched with AP1, ETS motifs (Fig.2g, two h)", yet the cluster 3 shows increased loss of accessibility in the mutant respect to the wt Sall4. The authors should clarify this discrepancy observed across panels 2G, H.

R4: We appreciate this comment very much. To make that clear, we add a new motif enrichment result related to ATAC-seq data (Figure 2h). We also revised the text into : "A total of 9344 ATAC-seq peaks can be classified into 6 clusters according to chromatin accessibility (Fig.2f). Among the 6 clusters, more than 2/3 of regions are open in Sall4^{K5A} but closed in Sall4^{WT} (C4, n=4573 and C6, n=1904). Besides, there are two interesting clusters that exhibit a loss of accessibility in Sall4^{K5A} but become accessible (C5) and inaccessible (C3) progressively in Sall4^{WT}, respectively. Examining the expression for genes whose promoter located within ATAC-seq peaks in each cluster indicates that patterns in transcription match those of chromatin accessibility (Fig. 2g). For regions that are more accessible in Sall4^{WT}, we observed a higher level of gene expression in Sall4^{WT} than in Sall4^{K5A} such as Mas1, Peg10, and Pkd1. In contrast, for regions in which accessibility is established in Sall4^{K5A} but remains inaccessible in Sall4^{WT}, there was a more significant increase in gene expression in Sall4^{K5A} than in Sall4^{WT} such as Bicc1, Fmo1, and Sox5 (Supplementary Fig.3i, j). Gene ontology analysis of genes associated with distinct clusters showed that the C4 and C6 loci correspond to those related to somatic cell maintenance and differentiation (e.g. regulation of mesenchymal stem cell differentiation), while the C5 loci are associated with cell cycle phase transition (Supplementary Fig.3k).

Motif enrichment for each cluster shows that enriched motifs are quite different between Sall4^{WT} and Sall4^{K5A}. For example, motifs from ETS (ETS1) and HOMEBOX family (Lhx3, Lhx1, Dlx1, Dlx3) members are specifically enriched in C6 and C4, respectively. Motifs for ETS and FOX family (FoxK2, FoxO3) members are both found in C6 and C4. Moreover, motifs for TFs from the AP-1 family such as Fosl2, Fra1/2, c-Jun, and JunB are also present in C6 (Fig.2h). These results are entirely consistent with our earlier findings that somatic gene loci enriched with somatic state specific TFs from AP-1 and ETS family members are barriers for reprogramming¹⁶⁻¹⁸. Several TFs

have already been shown significantly inhibits iPSC induction such as FoxK2 and FoxO319. Lhx3 and Dlx1/2 selectively drive fibroblast to distinct subtypes of neurons²⁰⁻²³. On the other hand, motif enrichment for C5 shows that pluripotent TFs such as OCT4/6, KLF4, and SOX17/21 are only found in Sall4 WT but not in K5A. These results suggest that the interaction between Sall4 and NuRD complex is required to reconfigure the chromatin architecture necessary for reprogramming.”

5. In Figure 3, the authors should show gene tracks of genes identified in Figure 3B (differentially acetylated regions).

R5: We have revised these figures as suggested by the reviewers. The gene tracks of the genes identified in Figure 3B have been added to Figure 3f and Figure S4c.

6. To support the conclusions in Figure 4, the authors should include IP-mass spec data for JDP2, JDP2(N12), ESRRB, and ESRRB (N12). The expectation is that only the JDP2(N12) protein gains interaction with NuRD components. Moreover, the authors should compare the JDP2 and JDP2(N12) ChIP-seq with GATAd2B ChIP-seq data.

R6: We thank the reviewers for their insightful suggestions and agree that IP-MS was helpful in demonstrating that Jdp2^{N12} rescues the Sall4^{K5A} defect during reprogramming. Therefore, we performed Jdp2N12 and Jdp2WT specific IP-MS during JGES reprogramming. Proteomic data showed that most NuRD subunits were significantly enriched only by Jdp2^{N12} but not by Jdp2^{WT}, suggesting that the NuRD complex may be a partner of Jdp2^{N12} that exerts gene regulatory activities (Fig. 5c).

However, we did not perform Esrrb IP-MS because Esrrb^{N12} could not rescue Sall4^{K5A} defect during reprogramming. According to motif enrichment by CUT&Tag analysis, ESRRB shows its genomic occupancy to pluripotent loci (Fig S5a). In contrast, CUT& Tag analysis shows that Jdp2 targets genomic loci enriched for somatic TFs such as Fra1, Fos, and Jun-AP1, indicating that Jdp2 acts as a repressor to close chromatin (Fig S5a). So Jdp2 may benefit from N12 grafting, but Esrrb does not. Therefore, we decided not to perform Esrrb-IP-MS experiments. On the contrary to N12-Jdp2, N12-Esrrb should occupy pluripotent loci and repress their expression, thus, inhibiting reprogramming, rather than promoting.

To validate Jdp2N12 may recruit NuRD complex to orchestrate chromatin remodeling and trigger somatic program inactivation. we performed GATAD2B CUT&Tag experiment during Jdp2^{N12} and Jdp2^{WT} reprogramming in combination with Esrrb, Glis1, and Sall4^{K5A} on day1. First, we categorized the JDP2^{N12} and JDP2^{WT} CUT&Tag peaks into the simplest tier of JDP2-Common, JDP2^{N12} specific, and JDP2^{WT} specific (Fig 4d), and then we analysed GATAD2B binding density for above three regions during Jdp2^{WT} and Jdp2^{N12} reprogramming (Fig.4j). For JDP2^{N12}-specific regions, where we observed higher GATAD2B binding density in Jdp2^{N12} reprogramming. Conversely, the JDP2^{WT}-specific regions exhibit increase GATAD2B occupancy in Jdp2^{WT} reprogramming. Collectively, these results suggest that grafting the NuRD interacting motif onto Jdp2 enable the synthetic factor to rescue reprogramming by establishing a N12-Jdp2-NuRD axis to partially replace the disrupted Sall4-NuRD axis.

Minor points:

- Figures 1F and 1H appear in the text in the wrong order.

We thank the reviewer for pointing out numbers that are out of order or incorrectly referenced in numbers. We switched Fig 1f and 1g and moved Fig1h to Fig S2g, and fixed the figure numbers in text.

- Figure 1F lacks the ESCs data.

We added the ESCs data in revised Fig 1g.

- Figure S1F and S1I show the same image for iCD3 and 7F conditions.

We thank the reviewer for pointing this out, we changed the image in Fig S1i for 7F with a new image.

- The authors should include viability data for MEFs depleted for Gatad2b/2a and CHD4.

To exam whether Gatad2a/2b and Chd4 depletion compromised MEFs viability, equal number of MEFs were seeded and infected by retrovirus shRNA for 2 days and another 2 days for puromycin selection, MEFs showed similar morphology and comparable cell number at day4 (Extended. Fig 3a, b). Together, these results suggesting that depletion of NuRD subunits have minimal effect for for the viability for MEFs.

- Figure 4C is missing the reprogramming data with the wt factors.

We agree and have updated the new figure with JGES reprogramming results.

- The description of the karyotype analysis in the methods section is unclear.

The requested method for karyotype analysis has been added to the method.

- The legend of FigS5 is unclear.

We have made that change.

- Different fonts are used across the text and Figures (Arial and Times). The authors should try to be consistent.

We apologize for this oversight and have changed fonts in the figures and text to Arial format in the revised version.

REVIEWER COMMENTS

Reviewer #1 (Remarks to the Author):

The authors address my previous points with a lot of data and analyses. They certainly have fixed a lot of issues and made it a better manuscript. I am not convinced that these 16 genes are really the thing. I think there is too much of using the data to fit a pre-existing hypothesis, and what they have shown clearly in all of their additional work is that there are very many ways to impede reprogramming. Which makes the effects seen with some of these 16 genes less impressive. I feel that the manuscript is a very detailed description of how the cells fail to reprogramme in different situations, but it is rare that we can actually assign causation to one specific protein or complex, even if correlation is very strong. For example:

Line 114: "NuRD mediates chromatin closing of somatic loci" In shGatad2b or shChd4 cells there is a failure of reprogramming. If we see that certain loci are not closed down properly, is that due to the general failure of reprogramming, or is it a direct consequence of NuRD not being there to do the job? It is not possible to discern which of these two possibilities is correct from the current experiments. Yes there is a high degree of correlation between loss of NuRD components and failure to close down specific chromatin regions...but there is no proof that NuRD does it.

Line 198. "Sall4 recruits NuRD complex to remove H3K27ac and close somatic loci". This is not shown. What the authors show is that in the S5A mutant there is less binding of Gatad2b at these sites as well as increased H3K27Ac. It is impossible to know whether the lack of Gatad2b causes the increase in H3K27Ac vs whether a change in the chromatin at these loci results in increases of H3K27Ac and reduced Gatad2b binding. There is no causality shown here, other than that this is caused by cells undergoing reprogramming when they are overexpressing Sall4 S5A.

The text, titles and conclusions should be toned down to reflect this and to remove inaccurate conclusion of direct function when what is shown is correlation.

I am very surprised that knockdown of Chd4 does not cause cell cycle arrest. The authors show their shRNA causes reduction in Chd4 transcript levels, and no apparent change in cell cycle parameters within 4 days. We do not know whether the protein is actually reduced, however. The fact that Chd4 deletion is lethal in mouse ES cells, and that homozygous loss of Chd4 is extremely rare in cancer indicate that few, if any cells can tolerate complete loss of Chd4. I cannot point to direct evidence that MEFs are one such cell type. That said, it may be that Chd4, if really dispensable in MEFs, becomes increasingly important as cells regress back to the ES cell state (where it is unquestionably essential) and that reprogramming is thus impossible without it for this reason. In contrast I expect that Gatad2b KO would be viable as presumably Gatad2a can compensate.

The elephant in the room is why this study comes to conclusions almost directly opposed to those reported by Mor et al. 2018 who argued that knocking down Chd4 or Gatad2a greatly facilitated reprogramming. Further, these authors argued that knocking down Chd4 in MEFs prior to reprogramming hindered the process due to a proliferation defect in the MEFs. The authors should at very least comment on the differences between the findings of these two studies and indicate why their results are so different from those in the literature.

Minor point:

Fig S2D-E. I don't understand what the legends mean

Reviewer #2 (Remarks to the Author):

Wang et al. have developed an efficient somatic cell reprogramming method by introducing Jdp1, Glis1, Esrrb and Sall4 (JGES) into MEF with the optimal chemically defined medium and suggested that Sall4-NuRD interaction is critical for closing somatic-specific open chromatin regions in the early phase of the reprogramming. This manuscript has been very much improved. However, the authors

should address the following points satisfactorily before this manuscript is accepted for publication in Nature Communications.

1. Line 111-112 "In combination, these results suggest that Sall4 may mediate reprogramming through components of the NuRD complex."

With the description in this location, it is unreasonable to indicate that Sall4 functions in the reprogramming through the NuRD complex. Moreover, the authors should compare the RNAseq data between JGE (Fig 1b) and NuRD complex KD (Fig. 1f) to make clear the relationship of the Sall4-NuRD axis at the molecular level.

2. Line 148-154 "To determine their contribution to JGES reprogramming, we made mutation or deletion as detailed in Supplementary Fig. 3f. Consistent with point mutation data described above, deleting N12 abolishes Sall4-dependent reprogramming, while other mutations have either no effect (ZF1) or limited impacts (C420A, ZFC2-4) (Supplementary Fig.3g, h), confirming that Sall4 mediates reprogramming primarily through its N12 domain that engages NuRD and secondarily through ZFC2-4 that likely engages the above mentioned factors involved in DNA repairs. As such, we continued to focus on the Sall4-NuRD complex in this study."

The authors should clearly state the basis for suggesting that ZFC2-4 are involved in DNA repairs and that ZFC2-4 are not involved in the NuRD complex.

3. Line216-219 "On the other hand, reduced level of H3K27ac and chromatin accessibility were found in Sall4K5A at several pluripotent gene loci (Sox2, Tead4, and Wnt6) (Supplementary Fig.4a, b, c), suggesting that certain chromatin regions may be opened and activated during this process."

Interesting observation. Does the NuRD complex bind these regions?

It would be nice to compare the results of GATAD2B cut&tag experiments by combining Fig3c and Fig3l. If NuRD is not binding, what is the reason for this? If it is, why the acetylation level drops in K5A mutated cells? It is worth discussing.

4. Line281-282 "These results suggest that closing of chromatin is accompanied by partial recruitment of NuRD complex".

It is unsure what the authors mean by "partial recruitment of NuRD complex".

5. Line285 "We then focused on genes that fail to be downregulated in G3 and abnormally activated in G6 during Sall4K5A reprogramming (Fig.2i)."

Fig.2i is missing.

6.

In Fig. S3a and S3b, the data of MEFs and ESCs should be presented in PCA as in the other figures.

7. In Extended Figure 2, the authors suggested that shNuRD treatments have little effect on morphological and pluripotent gene expression. However, these observations are inconsistent with previous reports (10.1038/ncb1372, 10.1074/jbc.M116.770248, etc.). The authors should comment on this discrepancy.

Reviewer #3 (Remarks to the Author):

The authors have addressed most of my comments, and the manuscript has improved tremendously.

I only have a couple of minor comments:

- the FACS analysis in Extended figure 1 (Thy-1 vs. SSEA-1) should present high Thy-1 cells that become gradually Thy-1 negative and SSEA-1 positive during reprogramming. Yet, there are no Thy-1 positive cells at Day 0 of reprogramming. This result is confusing as MEFs are Thy-1+ (~80%), and many labs have used this marker to follow reprogramming trajectories.

- The summary figure is convoluted, and it would be great if the authors could simplify their model in Figure 4 for the general audience.

Dear editor and reviewers, thank you for reviewing our manuscript and for the constructive comments, which greatly helped us to improve the manuscript. We have performed additional our experiments as suggested by all the reviewers, and also added additional literature to answer all the questions raised. The manuscript now has been carefully revised and the point-by-point responses provided below. We hope that your comments have been addressed accurately. Please note that the revised texts in manuscript are marked with **red** color and the responses presented in **blue** text.

Main changes and additions made to Figures.

1 We added Fig S2h, S2g, S2h, S2i to revised Figure S2. The original Fig S2f, S2g were moved to Fig S2j and S2k.

2 We added MEFs and ESCs samples to FigS3a and Fig3b in revised Fig S3.

3 We change the original Fig 3f to a new one in revised Fig3f.

4 The original Fig S4c was changed to a new one in revised FigS4c.

5 The original Fig 4j was replaced a new one in revised Fig 4j.

6 We added FigS5f and S5g to the revised Fig S5.

Reviewer #1 (Remarks to the Author):

The authors address my previous points with a lot of data and analyses. They certainly have fixed a lot of issues and made it a better manuscript. I am not convinced that these 16 genes are really the thing. I think there is too much of using the data to fit a pre-existing hypothesis, and what they have shown clearly in all of their additional work is that there are very many ways to impede reprogramming. Which makes the effects seen with some of these 16 genes less impressive.

I feel that the manuscript is a very detailed description of how the cells fail to reprogramme in different situations, but it is rare that we can actually assign causation to one specific protein or complex, even if correlation is very strong. For example:

Line 114: "NuRD mediates chromatin closing of somatic loci" In shGatad2b or shChd4 cells there is a failure of reprogramming. If we see that certain loci are not closed down properly, is that due to the general failure of reprogramming, or is it a direct consequence of NuRD not being there to do the job? It is not possible to discern which of these two possibilities is correct from the current experiments.

Yes there is a high degree of correlation between loss of NuRD components and failure to close down specific chromatin regions...but there is no proof that NuRD does it.

We appreciate the reviewer's insightful suggestion and agree that there is no direct evidence that knocking down NuRD leads to reprogramming failure. As a result, we hope that further analysis of the RNA-seq and ATAC-seq datasets, as well as the references listed below, will significantly support our argument.

First, previous studies of gene expression during reprogramming revealed that the loss of somatic cell identity occurs prior to the acquisition of pluripotency. Cells that become resistant to reprogramming have an increased expression of MEF-specific genes or exhibit abnormal gene expression patterns during reprogramming. Kathrin Plath and colleague reported somatic

specific genes linked to extracellular matrix were gradually downregulated during reprogramming (PMID: 28111071). We and others found IFN-signaling impedes the final transition to chimera-competent pluripotency (PMID: 30772174, 27884981). Many MEF-enriched somatic genes were downregulated at a slower rate by Gatad2b/Chd4 knockdown than by shLuciferase knockdown (Fig1F). According to Gene Ontology analysis, those genes are enriched for interferon-beta and extracellular matrix-related terms, which is consistent with previous research indicating that the aforementioned genes are barriers to somatic cell reprogramming. These findings imply that the NuRD complex may carry out the beneficial effects by inactivation of somatic specific program on JGES reprogramming.

Second, we discovered a similar transcriptome dynamic between JGES reprogramming by Gatad2b/Chd4/Luciferase knockdown and Sall4 dropout (JGE) reprogramming when we performed an integrated analysis for the RNA-seq data of knocking-down NuRD and Sall4 withdraw (below). During JGES reprogramming, somatic-specific genes in groups 1 and 5 are gradually inactivated (Fig1F), which cannot be silenced after knocking down Gatad2b or withdrawing Sall4, indicated a strong correlation between Gatad2b and Sall4. Similar relationships can also be found during pluripotency gene activation (group 4 and 6). These findings suggest that Sall4 may work with the NuRD complex to regulate MEF-specific genes.

Third, we wanted to see if there was any correlation between shGatad2b/shChd4 and chromatin accessibility, gene expression, and Sall4 occupancy at each OC/OC category. To begin, we counted the number of genes whose gene bodies or promoters were found in the ATAC-seq peaks region. In line with the results of chromatin accessibility, Gatad2b or Chd4 knockdown enriched hundreds of additional genes when compared to the control. Interestingly, when compared to the common and shLuciferase parts, Gatad2b knockdown enriched fewer genes from OC1 and OC2, but found more genes in OC3-OC6. The Chd4 knockdown reprogramming yielded similar results (Supplementary Fig.2f, g). These findings suggest that there is a delay in the inaccessibility of many somatic gene regions during reprogramming by Gatad2b/Chd4 knockdown.

Furthermore, when the occupancy of Sall4, Gatad2b, and H3K27ac was examined at OC regions, Sall4 and Gatad2b showed a higher binding density at shLuciferase than shGatad2b and shChd4 specific regions. However, Gatad2b or Chd4 knockdown resulted in an increased H3K27ac signal (Supplementary Fig.2h, i). These findings support the notion that the NuRD complex is involved in the early inactivation of the somatic program during JGES reprogramming.

Line 198. "Sall4 recruits NuRD complex to remove H3K27ac and close somatic loci". This is not shown. What the authors show is that in the S5A mutant there is less binding of Gatad2b at these sites as well as increased H3K27Ac. It is impossible to know whether the lack of Gatad2b causes the increase in H3K27Ac vs whether a change in the chromatin at these loci results in increases of H3K27Ac and reduced Gatad2b binding. There is no causality shown here, other than that this is caused by cells undergoing reprogramming when they are overexpressing Sall4 S5A. The text, titles and conclusions should be toned down to reflect this and to remove inaccurate conclusion of direct function when what is shown is correlation.

We appreciate the reviewer's thoughtful suggestion and agree that because Gatad2b is not a histone deacetylase, it cannot be responsible for the increase in H3K27ac at somatic sites. We use neutral and precise language in the revised manuscript to avoid misunderstandings about the decreased Gatad2b binding and increased H3K27ac.

I am very surprised that knockdown of Chd4 does not cause cell cycle arrest. The authors show their shRNA causes reduction in Chd4 transcript levels, and no apparent change in cell cycle parameters within 4 days. We do not know whether the protein is actually reduced, however. The fact that Chd4 deletion is lethal in mouse ES cells, and that homozygous loss of Chd4 is extremely rare in cancer indicate that few, if any cells can tolerate complete loss of Chd4. I cannot point to direct evidence that MEFs are one such cell type. That said, it may be that Chd4, if really dispensable in MEFs, becomes increasingly important as cells regress back to the ES cell state (where it is unquestionably essential) and that reprogramming is thus impossible without it for this reason. In contrast I expect that Gatad2b KO would be viable as presumably Gatad2a can compensate.

We appreciate the reviewer for insightful comment. To address the reviewer's concern about protein level and cell cycle arrest in Chd4 depleted ESCs, we did additional experiment to address this question. We performed Chd4 knockdown experiment in ESCs and verified knockdown efficiency by q-PCR and western blot analysis (Fig B, C, below). We did not observe apparent change in morphology and cell growth within 4 days. At passage 2 and after, Chd4-deficient ESCs, however, were smaller and grew more slowly than control cells (Fig A, below). When other researchers (Zhao et al., 2017, Mor et al., 2018) used inducible shRNA or siRNA to

achieve fast Chd4 knocking down at mRNA and protein levels, our slow lentivirus administration caused the neglected phenotype at day 4.

A second concern relates to the requirement of Chd4 during reprogramming. We observed minimal effect on somatic cell proliferation by retrovirus encoding shRNA specific for Chd4. As showed in Fig S2a, around 10%-20% of Chd4 remain in MEFs, which may help MEFs escape cell cycle pressure. In our reprogramming procedure, MEFs were infected with retrovirus encoding JGES and shRNA for 48 hours and then started reprogramming by induction medium. Similarly, Mor and colleagues discovered that siRNA-mediated knockdown of Chd4 performed before OSKM induction (pre-DOX) had a negative effect on reprogramming (Mor et al., 2018). As the reviewer mentioned and many literatures reported that Chd4 can function outside of NuRD complex and play important roles in cellular processes such as DNA-damage response and cell cycle progression, as well as 3D chromatin organization (Hou et al., 2020, Polo et al., 2010, Han et al., 2021). Moreover, several genome-wide localization studies have found that Chd4 also to be associated with large number of active gene loci in different cell types (Reynolds et al.2012, Whyte et al., 2012, Zhang et al., 2012, Williams et al., 2004). So, in our work, we do not have enough results to distinguish which molecular mechanism behind the multifaceted role of Chd4 during reprogramming.

The elephant in the room is why this study comes to conclusions almost directly opposed to those reported by Mor et al. 2018 who argued that knocking down Chd4 or Gatad2a greatly facilitated reprogramming. Further, these authors argued that knocking down Chd4 in MEFs prior to reprogramming hindered the process due to a proliferation defect in the MEFs. The authors should at very least comment on the differences between the findings of these two studies and indicate why their results are so different from those in the literature.

We appreciate the reviewer's thoughtful suggestion. We have added the following sentences to the discussion section:

Our findings appear to contradict multiple earlier studies that implicated NuRD subunits as a negative rheostat in reprogramming, including one that found that Gatad2a and Chd4 depletion resulted in up to 100% iPSC derivation efficiency (Mor et al., 2018). Our study is different from these other studies in a number of ways, including the reprogramming cocktail we used and the reprogramming conditions. Mor and colleagues used knockdown experiments by siRNA or knockout to inhibit Gatad2b or Chd4 during reprogramming, in contrast to our retrovirus delivery. While in our reprogramming system, the MEFs were infected with JGES retrovirus for reprogramming, they opted for transgenic "secondary reprogramming" embryonic fibroblasts (MEFs) that carry TetOn-inducible OKSM for iPSC induction. Furthermore, Mor and colleagues discovered that repressing Mbd3 and Chd4 with targeted siRNA prior to OKSM induction hampered the reprogramming process. Conversely, Santos and colleagues reported a positive role for MBD3/NuRD in transcription factor-mediated reprogramming of neural stem cells and epiblast stem cells to naive stem cells, implying a context-dependent role for the NuRD complex in pluripotency induction (Santos et al., 2014). Besides, we have reported that MEFs induced with OKSM and 7F (Jdp2, Esrrb, Sall4, Nanog, Kdm2b, Mkk6, Gkis1) follow distinct molecular trajectory during 7-day course to arrive final naïve state (Wang et al., 2019). Mor and colleagues proposed a model that Gatad2a/Mbd3 represses the same genes that OKSM try to reactivate. Given our unique system of reprogramming mediated by JGES, it is difficult to reconcile these divergent roles reported so far including our own. Our results thus may provide additional rationale to further investigate the role of NuRD components in various form of cell fate decisions, especially those involved in somatic cell reprogramming.

Minor point:

Fig S2D-E. I don't understand what the legends mean

We thank the reviewer for pointing this out. We have revised the figure legends as follows: Bar plot showing the number of common and specific peaks in each OC/CO category when comparing shGatad2b (d) or shChd4 (e) to shLuciferase. Colors indicated peaks found in shGatad2b or shChd4 (yellow) only, shLuciferase (blue) only, and both (red), respectively. Data under each column are presented as the percentage of common peaks found in shGatad2b or shChd4 relative to shLuciferase.

Reviewer #2 (Remarks to the Author):

Wang et al. have developed an efficient somatic cell reprogramming method by introducing Jdp1, Glis1, Esrrb and Sall4 (JGES) into MEF with the optimal chemically defined medium and suggested that Sall4-NuRD interaction is critical for closing somatic-specific open chromatin regions in the early phase of the reprogramming. This manuscript has been very much improved. However, the authors should address the following points satisfactorily before this manuscript is accepted for publication in Nature Communications.

1. Line 111-112 "In combination, these results suggest that Sall4 may mediate reprogramming through components of the NuRD complex."

With the description in this location, it is unreasonable to indicate that Sall4 functions in the reprogramming through the NuRD complex. Moreover, the authors should

compare the RNAseq data between JGE (Fig 1b) and NuRD complex KD (Fig. 1f) to make clear the relationship of the Sall4-NuRD axis at the molecular level.

We are grateful to the reviewer for these constructive comments and agree with that it was unprecise to make this conclusion. So, we accepted the reviewer's suggestion and compared the RNA-seq data between JGE and NuRD complex knockdown to make it clear.

By comparing the transcriptome dynamics of genes from those 6 groups between JGES reprogramming by NuRD knockdown and JGE reprogramming, we found a similar dynamic between JGE (orange) and shGatad2b (Red), indicating correlation between NuRD complex and Sall4 when reprogramming cells become pluripotent. In addition, we also observed that genes in group 1 and 5 that were inactivated progressively after JGES induction, showed higher normalized counts by Gatad2b knockdown than shChd4 and shLuciferase. The expression patterns in JGE were quite similar to the patterns by Gatad2b knockdown in group 5. Moreover, Sall4 dropout from JGES cocktail leads to a defect in the repression of somatic genes in group 1. Therefore, either NuRD complex knockdown or Sall4 dropout impair JGES reprogramming, and proteomics and transcriptome datasets imply Sall4 may bridge NuRD complex to repress somatic genes.

2. Line 148-154 “To determine their contribution to JGES reprogramming, we made mutation or deletion as detailed in Supplementary Fig. 3f. Consistent with point mutation data described above, deleting N12 abolishes Sall4-dependent reprogramming, while other mutations have either no effect (ZF1) or limited impacts (C420A, ZFC2-4) (Supplementary Fig.3g, h), confirming that Sall4 mediates reprogramming primarily through its N12 domain that engages NuRD and secondarily through ZFC2-4 that likely engages the above mentioned factors involved in DNA repairs. As such, we continued to focus on the Sall4-NuRD complex in this study.” The authors should clearly state the basis for suggesting that ZFC2-4 are involved in DNA repairs and that ZFC2-4 are not involved in the NuRD complex.

We appreciate the reviewer's insightful suggestion. We will clarify our views on the following points: First, our IP-MS results revealed that the common proteins enriched by Sall4 WT and K5A were involved in DNA damage repair. We can only conclude from the existing data that

ZFC2-4 are potential domains distributed throughout the SALL4 protein that are involved in DNA damage repair. Multiple studies, on the other hand, have found that a highly conserved N-terminal 12-amino acid motif in TFs like Sall4, Sall1, and Fog1 is both sufficient and required for NuRD recruitment (Lauberth et al., 2005, Miccio et al., 2010, Liu et al., 2018). Based on that, we come the conclusion that ZFC2-4 are likely involved in DNA damage repair but not involved in the recruitment of NuRD complex.

3. Line216-219 “On the other hand, reduced level of H3K27ac and chromatin accessibility were found in Sall4K5A at several pluripotent gene loci (Sox2, Tead4, and Wnt6) (Supplementary Fig.4a, b, c), suggesting that certain chromatin regions may be opened and activated during this process.”

Interesting observation. Does the NuRD complex bind these regions?

It would be nice to compare the results of GATAD2B cut&tag experiments by combining Fig3c and Fig3l. If NuRD is not binding, what is the reason for this? If it is, why the acetylation level drops in K5A mutated cells? It is worth discussing.

We appreciate the reviewer’s helpful comments and agree that a general analyses about the occupancy of cis-regulatory elements at pluripotent gene loci should be performed. Indeed, GATAD2B binds at above mentioned pluripotent gene loci (Sox2, Tead4, and Wnt6) in both WT and K5A reprogramming (Fig B). One may argue that the presence of Gatad2b at those gene loci in both WT and K5A condition, because K5A could not recruit NuRD. The distinct level of H3K27ac modification in K5A and WT cells may be due to the following reasons:

First, we believe the high level of H3K27ac in WT at those gene loci is a consequence of successful reprogramming. RNA-seq data showed that the expression of pluripotent gene Sox2 and Tead4 is higher in WT than K5A, which fits well with the observation of an increase level of H3K27ac in WT (Fig A). Previous studies indicated that the establishment of histone modification H3K27ac at actively transcribed pluripotent loci are closely associated with activation of pluripotent program, which is a stepwise process and dependent on cooperative binding of reprogramming factors. Therefore, Sall4 does not operate in isolation and need to work with other TFs to modify the local epigenetic environment. Thus, despite Gatad2b binding, Sall4 is necessary but not sufficient for the stepwise established H3K27ac modification at pluripotent gene loci, which validated by the results that single JGES dropout compromise reprogramming efficiency. So, the high level of H3K27ac at those gene loci in WT cells may be not dependent of Gatad2b occupancy. Another co-repressor complex with a major role in pluripotent gene activation during reprogramming is NCoR/SMRT complex, which contains HDAC3 but not HDAC1 and HDAC2 (Zhuang et al., 2018). In summary, high level of H3K27 at pluripotent gene loci in WT is characteristic of successful reprogramming, although we cannot exclude the participation of Gatad2b.

In contrast, the low level of H3K27ac at those gene loci in K5A may be associated with failure of reprogramming. Growing evidence suggested that the defective silence of somatic program compromise activation of pluripotent program (Li et al., 2017, Chronis et al., 2016). In this regard, the lower H3K27ac levels at these loci in K5A could be caused by failing to shutdown somatic programs during the early stages of reprogramming. Genomic view of ATAC-seq at Sox2 and Tead4 gene loci showed strong signal in WT and ES cells but remain low level in MEF and K5A cells (Fig C), suggesting K5A could not open those pluripotent gene loci. Therefore, Gatad2b binding at those gene loci is associated with its repressive function, as it does in MEFs.

4. Line281-282 “These results suggest that closing of chromatin is accompanied by partial recruitment of NuRD complex”. It is unsure what the authors mean by “partial recruitment of NuRD complex”.

We appreciate the reviewer for pointing this out.

We conducted an integrated analysis of Sall4 and H3K27ac CUT&Tag data in order to study the genome-wide correlation between sites bound by Sall4 and H3K27ac. As a result, we discovered 9828 gene loci with elevated levels of H3K27ac when Sall4 was unable to recruit the NuRD complex during reprogramming. When a similar analysis was done to look at the sites that Sall4 and Gatad2b both bound, we discovered that (7199+463) sites lost Gatad2b occupancy. However, the number of overlapping sites was reduced to 610 when we compared the sites bound by Gatad2b, Sall4, and H3K27ac, indicating that a significant fraction of those sites was not shared by Gatad2b and H3K27ac despite Sall4 binding. Though the motif enrichment showed the 610 sites are critical chromatin regions that should be closed during reprogramming, we could not exclude other mechanisms involved in it. We recognize the limitation of original sentence, so we revised the following sentence: These results suggest that the NuRD complex is associated with closing of chromatin.

5. Line285 “We then focused on genes that fail to be downregulated in G3 and abnormally activated in G6 during Sall4K5A reprogramming (Fig.2i).”

Fig.2i is missing.

Apologies for the missed figure, which have been corrected

6. In Fig. S3a and S3b, the data of MEFs and ESCs should be presented in PCA as in the other figures.

As suggested by the reviewer, we added MEFs and ESCs to the PCA.

7. In Extended Figure 2, the authors suggested that shNuRD treatments have little effect on morphological and pluripotent gene expression. However, these observations are inconsistent with previous reports (10.1038/ncb1372, 10.1074/jbc.M116.770248, etc.). The authors should comment on this discrepancy.

The reviewer raises a very pertinent point. Another reviewer made a similar observation regarding the phenotype of Chd4 depletion in ESCs. In fact, there was no discernible change in pluripotent gene expression or morphological at day 4 of our knockdown experiment. At passage 2 and after, Chd4-deficient ESCs, however, were smaller and grew more slowly than control cells (Fig A). When other researchers used inducible shRNA or siRNA to achieve fast Chd4 knocking down at mRNA and protein levels, our slow lentivirus administration caused the neglected phenotype at day 4 (Mor et al., 2018, Zhao et al., 2016).

We verified knockdown efficiency by quantifying Chd4 mRNA and protein levels, with knockdown efficiency around 50%. In contrast to jbc.M116.770248, etc. reported, we found the transcript levels of Oct4 decreased. Similar to their results, the expression levels of Gata6, Cdx2, T, and Hand1 increased to various degrees in Chd4 knockdown ESCs.

Reviewer #3 (Remarks to the Author):

The authors have addressed most of my comments, and the manuscript has improved tremendously.

I only have a couple of minor comments:

- the FACS analysis in Extended figure 1 (Thy-1 vs. SSEA-1) should present high Thy-1 cells that become gradually Thy-1 negative and SSEA-1 positive during reprogramming. Yet, there are no Thy-1 positive cells at Day 0 of reprogramming. This result is confusing as MEFs are Thy-1+ (~80%), and many labs have used this marker to follow reprogramming trajectories.

The reviewer raised a key point about dynamics of both THY1 and SSEA1 positive cells during reprogramming. To address the reviewer's concern, we purchased and tested three distinct THY1 antibodies and two different SSEA1 antibodies, as a result, validated antibodies were identified.

First, we chose MEFs and mESCs as control to test for Thy1 and SSEA1 antibodies, respectively. Flow cytometry showed that 83.7% of MEFs were Thy1-positive and 50.4% mESCs were SSEA1-positive, indicating that these two antibodies were reliable.

Second, as somatic cell reprogramming progresses, Thy1-positive cells gradually decline and rise slightly in the later stages, which is similar to previous reports (PMID: 35385732). Besides, the proportion of SSEA1-positive cells gradually increases from day1 to day7.

Importantly, knocking down Chd4, Gatad2a/2b resulted in an increase in the proportion of THY1-positive cells when compared to shLuciferase samples during JGES reprogramming. Similarly, knocking down the above three subunits of the NuRD complexes results in failure of SSEA1-positive cell activation. These results reveal NuRD complex is indispensable in JGES reprogramming system.

- The summary figure is convoluted, and it would be great if the authors could simplify

their model in Figure 4 for the general audience.

As suggested by the reviewer, we have simplified the working model in Figure 4.

REVIEWERS' COMMENTS

Reviewer #1 (Remarks to the Author):

The authors have addressed my concerns that they have drawn hard conclusions from what are correlative data by presenting us with several more correlations. So my concerns on this regard still stand.

I also still feel that they have analysed the data to fit a pre-existing hypothesis. That said they have carried out a large number of analyses and generate a large amount of data, with large scale sequencing all done in duplicate. I suspect this will be useful for the reprogramming community.

It appears that the reason the MEFs are still viable after CHD4 KD is because the level of knockdown is not very high, so that there is enough CHD4 to keep the cells alive. So the results they see are due to reduced CHD4 levels, not absence of CHD4.

And they have now dealt with the Mor et al. issue in a completely reasonable way.

Reviewer #1 (Remarks to the Author):

The authors have addressed my concerns that they have drawn hard conclusions from what are correlative data by presenting us with several more correlations. So my concerns on this regard still stand.

I also still feel that they have analysed the data to fit a pre-existing hypothesis. That said they have carried out a large number of analyses and generate a large amount of data, with large scale sequencing all done in duplicate. I suspect this will be useful for the reprogramming community.

It appears that the reason the MEFs are still viable after CHD4 KD is because the level of knockdown is not very high, so that there is enough CHD4 to keep the cells alive. So the results they see are due to reduced CHD4 levels, not absence of CHD4.

And they have now dealt with the Mor et al. issue in a completely reasonable way.

Much appreciate for the constructive comments, which greatly helped us to improve the manuscript. We are pleased that the reviewers show a positive response to our second-round revision. We performed an unbiased analysis of high-throughput sequencing data, and analyzed genes expression, chromatin accessibility dynamics and underlying molecular mechanisms in SALL4 WT and K5A reprogramming. Since NuRD is a conserved transcriptional corepressor complex, we focus more on gene silencing-related regulatory mechanisms during reprogramming. We and others have also found that the inactivation of somatic programs is a prerequisite for successful reprogramming. Indeed, our data also prove that somatic cell-related programs cannot be inactivated after Sall4 mutated.

Meanwhile, we realize that there is still some room for improvement in this manuscript. It could not completely delete CHD4 within cells by knockdown experiments, so solid conclusions about the survival of MEF cells or other cells is CHD4 dependent or not requires knockout experiment. Alternative methods such as PROTAC or auxin-induced degron (AID) system could also be feasible to delete CHD4.